# Natural xanthones as α-Mangostin induce vasorelaxation involving key gating residues in the S6 domain of BK channels

**Soenke Cordeiro[1], Robert Patejdl[2], Thomas Baukrowitz[1], Marianne A Musinszki[1]\***

[1]Institute of Physiology, Christian-Albrechts-University Kiel, Kiel, Germany;
[2]Department of Human Medicine, Health and Medical University Erfurt, Erfurt, Germany

## eLife Assessment

The present manuscript by Cordeiro et al. shows **convincing** evidence that α-mangostin, a xanthone obtained from the fruit of the Garcinia mangostana tree, behaves as a strong activator of the large-conductance (BK) potassium channels. The authors suggest that α-mangostin activation of the BK channel is state-independent, and molecular docking and mutagenesis suggest that α-mangostin binds to a site in the internal cavity. Additionally, the authors show that α-mangostin can relax arteries, further suggesting the plausibility of the proposed effects of this compound. These are **valuable** findings that should be of interest to channel biophysicists and physiologists alike.

**\*For correspondence:**
m.musinszki@physiologie.uni-kiel.de

**Competing interest:** The authors declare that no competing interests exist.

## Abstract
Polyphenolic compounds are widely explored for health benefits, including hypertension, but their active ingredients, molecular targets, and mechanisms remain poorly defined. We identify the xanthone Mangostin from *Garcinia mangostana* as a potent modulator of several potassium channels, with large-conductance $K^+$ (BK) channels as its primary target for vasorelaxation. Mangostin-activated BK channels as α subunits alone, in complexes with vascular β1 subunits, and in reconstituted BKα/β1–$Ca_v$ nanodomains. It shifted BK voltage activation to more negative potentials by antagonizing channel closure and promoting channel opening without markedly altering $Ca^{2+}$ sensitivity. Docking, competition, single-channel analysis, and mutagenesis localized the binding site in the pore cavity below the SF, involving gating-critical S6 residues I308, L312, and A316, and suggest that Mangostin stays bound in closed and open states. These findings establish BK channel activation as the core molecular mechanism driving Mangostin's vascular effects and define its structural mode of action, informing nutraceutical safety assessment and BK-targeted drug design.

## Introduction

Nutraceuticals (superfoods, dietary supplements, and traditional remedies) promise a plethora of health benefits that are usually not definitely proven by clinical studies. Well-known examples are resveratrol present in grapes and red wine (*Jang et al., 1997*; *Brown et al., 2024*), carotenes and tocopherols (*Alpha-Tocopherol and Beta Carotene Cancer Prevention Study Group, 1994*; *Hennekens et al., 1996*; *Xin et al., 2022*), or the recent resurgence of quercetin (*Lee et al., 2021*; *Di Pierro et al., 2022*). In evaluating clinical relevance and exploring pharmacological opportunities of such phytochemicals, obstacles are often the identification of the biologically active substance in complex matrices of plant secondary metabolites and the lack of a definite cellular determinant that could link a biologically active substance to a physiological process. However, a clear molecular target and mechanism of action that can explain its effects is the prerequisite to further drug development.

Natural polyphenols are known to modulate diverse potassium ($K^+$) channel families. For example, various members of the flavonoid subgroup activate $K_{Ca}$ (BK) channels as well as several $K_v$, $K_{ir}$, and $K_{2P}$ channels, including TREK-1 channels (*Gierten et al., 2008*; *Nardi and Olesen, 2008*; *Kim et al., 2011*; for a comprehensive review see *Richter-Laskowska et al., 2023*). A recent study suggested that α-Mangostin, a xanthone from *Garcinia mangostana,* modulates ion channels and binds in the pore cavity of TREK $K_{2P}$ channels (*Kim et al., 2023*). α- and γ-Mangostins are the main xanthones found in the mangosteen fruit pericarp, whose ethanolic extracts are used as traditional medicine and consumed as nutraceuticals. They are thought to exert a plethora of health-promoting effects, such as cardioprotective (*Eisvand et al., 2022*), analgesic (*Kim et al., 2023*), antioxidant (*Kong et al., 2022*), anti-inflammatory (*Kim, 2021*), antidiabetic (*John et al., 2022*), antifibrotic (*Li et al., 2019*), antimicrobial (*Tatiya-Aphiradee et al., 2016*), and antiproliferative effects (*Majdalawieh et al., 2024*). Interestingly, very recent research reported that α-Mangostin has antihypertensive properties, and it decreased systolic and diastolic blood pressure in a rat model (*Xu et al., 2024*). Earlier studies had demonstrated the relaxation of aortic rings upon Mangostin incubation, but remained somewhat contradictory as to the site of action and the molecular target (*Chairungsrilerd et al., 1996*; *Chairungsrilerd et al., 1997*; *Tep-Areenan and Suksamrarn, 2012*). Hence, the molecular mechanism of this antihypertensive effect of Mangostin remains unclear.

Major potassium channels present in vascular smooth muscle are $Ca^{2+}$-activated BK channels, also known as MaxiK, Slo1, or $K_{Ca}1.1$ channels (*Wu and Marx, 2010*; *Tykocki et al., 2017*; *Pereira da Silva et al., 2022*). They are synergistically activated by intracellular calcium ($Ca_i^{2+}$) and voltage, and they play a crucial role in the regulation of vascular smooth muscle tone, with implications for blood pressure regulation. BK channels act as feedback regulators, which counteract depolarization and balance the elevation of $Ca_i^{2+}$ through voltage-dependent L-type $Ca^{2+}$ channels. The pore-forming BKα subunit is expressed ubiquitously, and tissue-specific properties are conveyed by regulatory β-, γ-, and LINGO subunits (*Latorre et al., 2017*; *Gonzalez-Perez and Lingle, 2019*; *Dudem et al., 2020*). In vascular smooth muscle cells, BKα assembles with the β1 subunit, which enhances its apparent $Ca_i^{2+}$ sensitivity, decreases voltage sensitivity (*Dworetzky et al., 1996*; *Brenner et al., 2000*; *Orio and Latorre, 2005*; *Li and Yan, 2016*), and modulates its pharmacological properties (*McManus et al., 1995*; *Hanner et al., 1997*).

Structurally, BK channels are homotetramers consisting of the pore-forming α subunits with seven transmembrane segments (S0–S6), a linker region, and two intracellular RCK domains. The RCK domains and their interface to the transmembrane segments contain different $Ca_i^{2+}$ binding sites, S1–S4 comprise the voltage-sensing domain, and S5 and S6 together with the pore helix constitute the pore domain. Both depolarization or $Ca_i^{2+}$ elevation can open the pore via allosteric mechanisms, and this regulation, as well as the modulation by mutations, small molecules, and regulatory subunits, can be described well by an allosteric gating model (*Horrigan and Aldrich, 2002*; *Latorre et al., 2017*). However, the structural determinants that underlie this allosteric modulation remain unclear. The S6 segment has been shown to undergo conformational changes during activation, and several gating-sensitive residues, as well as modulator binding sites for molecules of surprisingly different chemical structure, are known in this pore region or the adjacent linker to the cytoplasmic C-terminus (*Nardi and Olesen, 2008*; *Zhou et al., 2011*; *Roy et al., 2012*; *Hoshi et al., 2013b*; *Chen et al., 2014*; *Webb et al., 2015*; *Hoshi and Heinemann, 2016*; *Schewe et al., 2019*; *Rockman et al., 2020*; *Gonzalez-Sanabria et al., 2025*). Investigations of the activation mechanism indicated that these substances act without directly affecting the voltage- or $Ca^{2+}$ sensory domains, probably by modifying the close interactions of S5 and S6 with the voltage sensor bundle and the gating ring that enable opening of the channel.

Small molecule BK channel activators are promising to treat diverse diseases like fragile X syndrome, overactive bladder, pulmonary hypertension, chronic obstruction, erectile dysfunction, and reperfusion injury; however, no prospective drug successfully completed a phase III trial yet (*Bentzen et al., 2014*; *Barenco-Marins et al., 2025*; *Ferraguto et al., 2024*). Therefore, adding a pharmacophore derived from the xanthone group may aid in developing future drug candidates.

We used a combination of functional studies in cultured cells and vascular tissue, as well as molecular docking, to show that BK channels are a molecular target of Mangostins and that they mediate vascular relaxation observed upon Mangostin application. BK channels were opened most prominently among the $K^+$ channel representatives tested and were activated by α-Mangostin as well as

by an extract of the mangosteen pericarp marketed as a nutraceutical. Investigation of the activation mechanism showed that α-Mangostin essentially facilitates BK channel activation by shifting its voltage dependence, modulating gating kinetics, and enhancing the open probability, without changing the $Ca_i^{2+}$ sensitivity. These effects were due to direct binding to the pore-forming BKα subunit. Competition experiments, molecular docking, and scanning mutagenesis identified the binding region in S6 just below the selectivity filter (SF). Mangostin activation was preserved in reconstructed nanodomains with $Ca_v$ channels, which cause local elevation of intracellular $Ca_i^{2+}$, mimicking the physiological situation in smooth muscle cells. Finally, we show that BK channels specifically mediate relaxation of aortic preparations from mice. Our study explains the molecular activation of BK channels by a natural xanthone and sheds light on the mechanism underlying one of its claimed health benefits.

## Results
### α-Mangostin particularly activates BKα and BKα/β1 channels

In previous studies, α-Mangostin was shown to activate members of the $K_{2P}$ channel family, while other $K^+$ channels were inhibited (*Kim, 2021*; *Kim et al., 2023*). We aimed to obtain a broader pharmacological profile of α-Mangostin in $K^+$ channels, and we investigated members of all six subfamilies

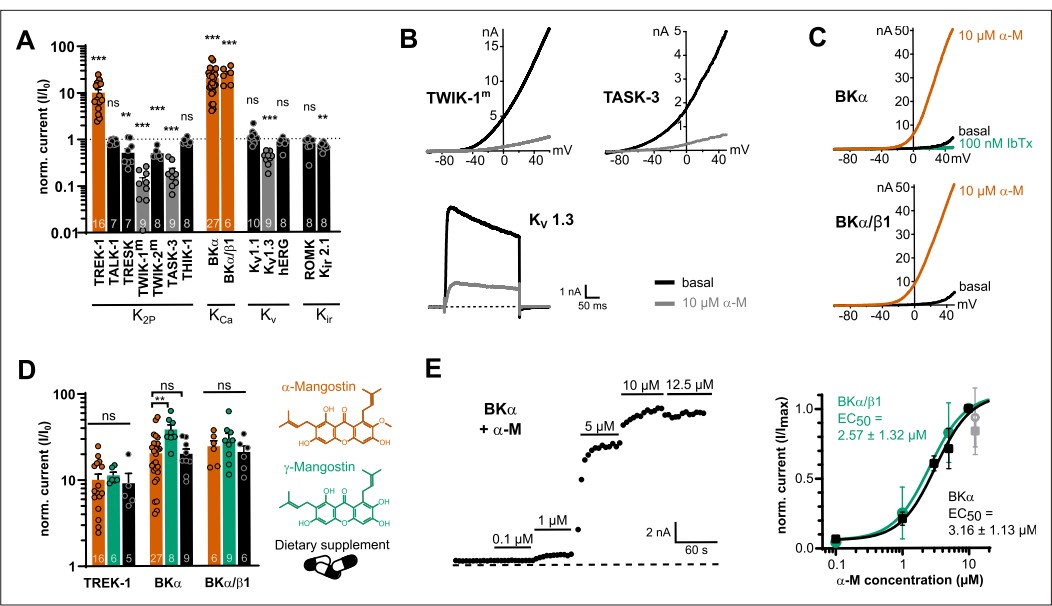

**Figure 1.** Mangostin potently activates BKα and BKα/β1 channels compared to other potassium channel representatives. (**A**) Current fold change ± SEM of currents of representatives from different potassium channel families upon application of 10 μM α-Mangostin. $K_{2P}$ and $K_{Ca}$ channel currents were recorded with ramp protocols from –100 to +50 mV and analyzed at +40 mV; $K_v$1.1 and $K_v$1.3 channel currents were evoked using a rectangle pulse to +40 mV, and hERG currents were recorded with a rectangle pulse to +60 mV followed by hyperpolarization to –120 mV, which was analyzed. $K_{ir}$ channels were recorded using ramp protocols from –150 to +50 mV, and currents were analyzed at –140 mV. All measurements were made in transiently transfected HEK293 cells in physiological potassium gradients with 100 nM intracellular free $Ca^{2+}$ with a holding potential of –80 mV. TWIK-1$^m$ and TWIK-2$^m$ denote channels where the retrieval motif was removed to improve membrane expression, and intracellular $K^+$ was exchanged for $Rb^+$ to enhance currents. (**B**) Representative current traces of channels that were inhibited more than 60% by 10 μM α-Mangostin. (**C**) Representative current traces of BKα and BKα/β1 channels activated by 10 μM α-Mangostin (α-M). (**D**) Current fold change ± SEM after application of α-Mangostin, γ-Mangostin, and a dietary supplement to TREK-1, BKα, and BKα/β1 channels. Currents were recorded and analyzed as in (**A**). (**E**) Representative time course of the dose-dependent activation of BKα channels by increasing concentrations of α-Mangostin (left) and the resulting dose–response relationships for BKα and BKα/β1 channels (right). Currents were recorded and analyzed as in (**A**); the grey data point at 12.5 μM was not included in the Hill fit. Data and statistics see Source data file 1.

The online version of this article includes the following figure supplement(s) for figure 1:

**Figure supplement 1.** Effect of α-Mangostin on the current of potassium channels from different families.

of $K_{2P}$ channels as well as representatives of the $K_v$ and $K_{ir}$ channel families. Confirming the previous studies, we observed a strong activation of TREK-1 $K_{2P}$ channels (fold change 10.33 ± 1.64; *Figure 1A*, *Figure 1—figure supplement 1*) and inhibition of TRESK $K_{2P}$ channels as well as $K_v1.3$ channels by ≈50% and ≈60%, respectively (*Figure 1A, B*, *Figure 1—figure supplement 1*). Furthermore, we identified three additional targets of α-Mangostin: strongly inhibited were the $K_{2P}$ members TWIK-$1^m$ (≈88%), TWIK-$2^m$ (≈50%), and TASK-3 (≈80%) (*Figure 1A, B*, *Figure 1—figure supplement 1*). In contrast, BK channels were most potently activated, with a fold change of 20.41 ± 2.39 for BKα channels and of 24.42 ± 3.85 for BKα/β1 channels, which is the complex predominantly present in smooth muscle cells of the vasculature (*Figure 1A, C*). This suggested that BK channels, especially the BKα/β1 heteromer, could be the molecular target mediating the vasorelaxant Mangostin effects reported in above studies.

The preparations from *G. mangostana* plants that are consumed as 'nutraceuticals' contain an unstandardized mixture of secondary plant metabolites, among them other Mangostin derivatives. Therefore, we also investigated their modulatory effect on TREK-1, BKα, and BKα/β1 currents (*Figure 1D*). The very similar γ-Mangostin-activated BKα channels with slightly higher efficacy (fold change of 38.54 ± 4.8) than TREK-1 and BKα/β1 channels (fold change of 11.35 ± 1 and 29.32 ± 5.03). Interestingly, the dietary supplement (solubilized equivalent to 10 µM α-Mangostin) activated TREK1, BKα, and BKα/β1 channels comparably to pure α-/γ-Mangostin (fold change of 9.18 ± 2.74, 19.98 ± 2.99, and 20.92 ± 3.49; *Figure 1D*), highlighting that mangostane xanthones retain their activity in such extracts.

The application of different α-Mangostin concentrations resulted in a fast and dose-dependent activation of BKα and BKα/β1 channels, with a similar apparent $EC_{50}$ of 3.16 ± 1.13 and 2.57 ± 1.32 µM, suggesting that the potency of α-Mangostin is not affected by the presence of the β1-subunit (*Figure 1E*). However, we frequently noticed a decline of currents at concentrations >12 µM and limited solubility of higher Mangostin concentrations, which may imply that the actual maximal activation could not be reached.

## α-Mangostin shifts the voltage activation of BKα and BKα/β1 channels to more negative values

Having established the prominent activation of BKα and BKα/β1 channels by α-Mangostin, we wanted to gain more insight into its activation mechanism. The open probability of BK channels is controlled by voltage signals, calcium signals, or shift of the closed–open equilibrium (e.g., by mutations or modulators), and the current further depends on the single-channel amplitude and the time the channel adopts an open conductive state. We therefore investigated these aspects of BK channel gating in macroscopic and single-channel measurements.

As we already observed in the ramp measurements, the threshold of BK voltage activation was shifted to more negative potentials upon α-Mangostin application (*Figure 1C*). Therefore, we first examined if α-Mangostin affects the voltage activation of BK channels. We used a step protocol over a range of potentials followed by a repolarization step to elicit tail currents in symmetrical bi-ionic conditions where intracellular $K^+$ was replaced with $Cs^+$ to reduce the strong outward currents in whole-cell experiments (*Figure 2A*; $Cs^+$ does not alter voltage activation *Piskorowski and Aldrich, 2006*). We quantified the change in voltage activation induced by α-Mangostin by calculating the voltage of half-maximal activation ($V_{1/2}$) from the Boltzmann fit of conductance–voltage (*GV*) relationships (*Figure 2B*). The application of 10 µM α-Mangostin caused a substantial shift of the $V_{1/2}$ by 53.08 ± 4.9 mV to more negative voltages in BKα channels (from 110.45 ± 2.69 to 57.37 ± 3.6 mV). This effect was even more pronounced in smooth muscle-like BKα/β1 channels, where $V_{1/2}$ shifted by 82.42 ± 4.96 mV to more hyperpolarized voltages (from 147.25 ± 5.66 to 64.83 ± 4.25 mV). The slope of the Boltzmann fit was not different before and after α-Mangostin activation in both channels (*Figure 2B*, insets). Hence, BKα and BKα/β1 channels already opened at less depolarized voltages in the presence of α-Mangostin, whereas voltage activation itself was not affected.

## α-Mangostin predominantly affects deactivation of BKα and BKα/β1 channels

An inspection of the currents elicited by the families of rectangle pulses shows that gating kinetics are altered in the α-Mangostin-activated state (*Figure 2A*). The shift in $V_{1/2}$ toward more negative

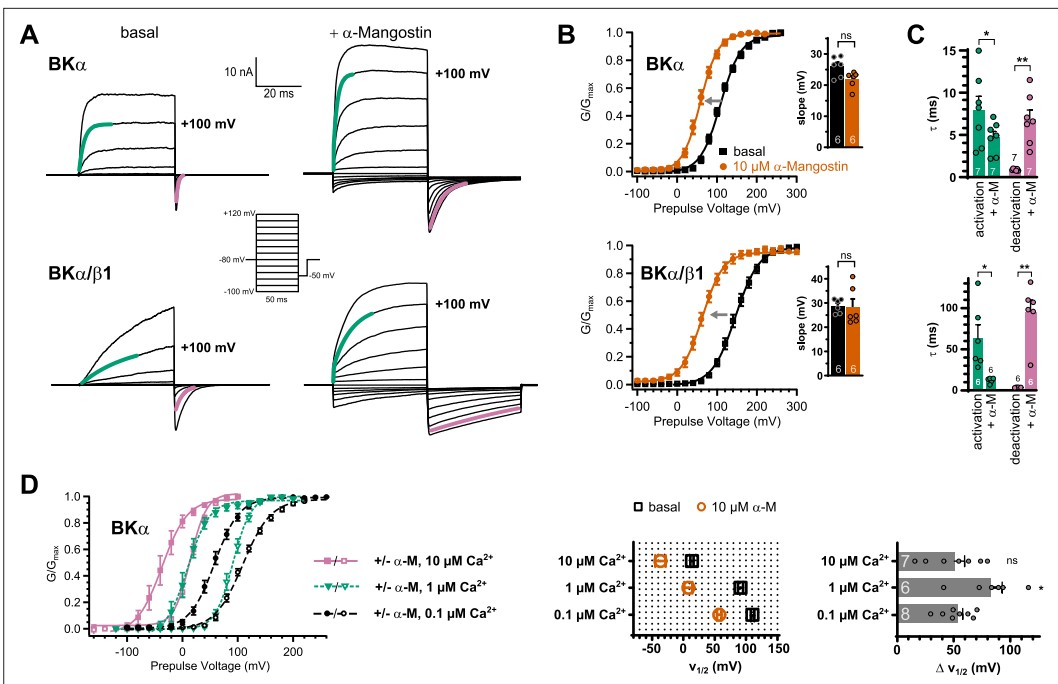

**Figure 2.** Effects of α-Mangostin on BKα and BKα/β1 channel gating. (**A**) Representative current traces for BKα and BKα/β1 channels before and after activation by 10 µM α-Mangostin. Cells were measured in symmetrical 140 mM bi-ionic conditions with 140 mM Cs$^+$ as the intracellular ion and 100 nM free Ca$_i^{2+}$. Currents were elicited by a family of rectangle pulses from –100 to up to +300 mV in 20 mV increments from a holding potential of –80 mV, followed by repolarization to –50 mV to elicit inward tail currents. (**B**) GV relationships for BKα and BKα/β1 channels in the basal state and activated by 10 µM α-Mangostin, derived from tail current analysis of recordings as in (**A**). The gray arrow illustrates the shift of voltage activation toward more negative voltages caused by α-Mangostin and the inset shows the slope ± SEM of the Boltzmann fits. (**C**) Activation and deactivation kinetics of BKα and BKα/β1 channels in the basal state and after activation by 10 µM α-Mangostin. The τ ± SEM of activation/deactivation was determined from exponential fits to the current traces at +100 mV, as shown by the green and purple colored lines in panel (**A**). (**D**) GV relationships of BKα channels in different free Ca$_i^{2+}$ concentrations before and after activation by 10 µM α-Mangostin, derived from tail current analysis as above. Pulse voltages were –100 to +300 mV/repolarization to –50 mV from a holding potential of –80 mV for 100 nM free Ca$_i^{2+}$, –120 to +200 mV/repolarization to –50 mV from a holding potential of –80 mV for 1 µM free Ca$_i^{2+}$, and –160 mV to +100 mV/repolarization to –80 mV from a holding potential of –120 mV for 10 µM free Ca$_i^{2+}$, in symmetrical 140 mM bi-ionic conditions with 140 mM Cs$^+$ as the intracellular ion. The $V_{1/2}$ values ± SEM before and after α-Mangostin application and the resulting shifts (Δ$V_{1/2}$ ± SEM) are shown in the bar graphs. Data and statistics see Source data file 1.

The online version of this article includes the following figure supplement(s) for figure 2:

**Figure supplement 1.** Change of activation and deactivation time course upon α-Mangostin activation of BKα channels measured with 10 µM Ca$_i^{2+}$.

---

potentials upon α-Mangostin activation could be caused either by an acceleration of channel activation, or by a slowing down of channel deactivation, or a combination of both. We therefore analyzed the kinetics of activation and deactivation by fitting monoexponential functions to the time course of outward and tail currents to obtain their time constants (τ). BKα channels are characterized by fast activating current and a fast deactivation visible as decay of the tail current, while the presence of the β1 subunit slows these kinetics down (**Figure 2A**, bottom; **Dworetzky et al., 1996**). After application of 10 µM α-Mangostin, activation of BKα channels was moderately accelerated 1.7-fold at +100 mV, with a τ of activation of 7.96 ± 1.64 ms before and 4.71 ± 0.71 ms afterwards (**Figure 2C**). Additional measurements in 10 µM Ca$_i^{2+}$ where outward currents are present at lower voltages showed a bigger effect with less depolarization (0–100 mV; **Figure 2—figure supplement 1A**). The τ for deactivation at –50 mV increased ≈7-fold from 0.9 ± 0.04 to 6.85 ± 1.11 ms at +100 mV (**Figure 2C**; **Figure 2—figure supplement 1B**). Again, α-Mangostin action was more pronounced in the smooth muscle-like BKα/

β1 channels, where activation was more than fivefold faster with $\tau$ values of 63.76 ± 16.03 ms before and 12.36 ± 1.20 ms after α-Mangostin application. Deactivation kinetics were most affected with a substantial ≈27-fold increase in $\tau$ from 3.60 ± 0.16 to 95.6 ± 13.99 ms (*Figure 2C*). These results suggest that the mechanism of α-Mangostin action differentially affects gating transitions that are associated with activation and deactivation.

As BK channels can be massively activated by $Ca_i^{2+}$ under physiological conditions, we further obtained *GV* relationships in the basal and in the activated state for two higher $Ca_i^{2+}$ concentrations relevant in smooth muscle cells (*Figure 2D*). α-Mangostin shifted the $V_{1/2}$ in 10 µM free $Ca_i^{2+}$ (representative for a local increase subsequent to a $Ca^{2+}$ spark) by 50.55 ± 9.13 mV to more negative potentials, comparable to the resting state with 0.1 µM free $Ca_i^{2+}$ (53.08 ± 4.9 mV), and a slightly higher $V_{1/2}$ shift of 82.74 ± 10.17 mV in 1 µM $Ca_i^{2+}$ (a range more likely in a global $Ca_i^{2+}$ elevation). Thus, α-Mangostin activates BK channels across a wide range of intracellular $Ca^{2+}$ concentrations, indicating $Ca^{2+}$-independent efficacy.

## α-Mangostin activation mechanism on single-channel and macroscopic current level

The marked slowing of deactivation visible as the slow tail current decay in macroscopic current recordings suggests that individual channels are open for longer periods of time in the presence of α-Mangostin. To investigate the effect of α-Mangostin on individual channels, we recorded single-channel currents in excised inside-out patches from HEK293 cells in the presence of 100 nM free $Ca_i^{2+}$ (*Figure 3A*). Under these conditions, the open probability ($P_o$) of BK channels at a potential of +40 mV was very low (0.002 ± 0.0008), but rose to 0.77 ± 0.08 after activation by 10 µM α-Mangostin (*Figure 3B*) with only very brief closings (*Figure 3A*, 1 s inset). The single-channel amplitude derived from the all-points-histogram was not different between basal and α-Mangostin-activated states (9.18 ± 0.29 and 9.76 ± 0.26 pA; *Figure 3B*). We constructed dwell time distributions to determine the differences in the duration that a single channel resides in the closed or open state (*Figure 3C*). The log closed dwell time distributions showed a large shift of the main component to shorter dwell times, from 2.38 ± 0.23 (237 ms) to –1.39 ± 0.36 (less than 0.1 ms). The open dwell time distribution revealed that the log open time was markedly reduced from –0.84 ± 0.45 to 0.8 ± 0.24 (i.e., from 0.14 to 6.28 ms).

BK channels are known to show bursting gating behavior (*Geng and Magleby, 2014*). As bursts did not occur in our measurements in resting conditions, we elevated free $Ca_i^{2+}$ to be able to perform burst analysis (*Figure 3D*). Mean burst duration was increased and the long closures between bursts were shortened, explaining the increase in overall open probability (*Figure 3E*). The intraburst kinetic distributions showed that α-Mangostin increased intraburst closed time from 20.8 ± 1.2 to 29.4 ± 1.3 ms, thereby reducing very brief closures (flicker), indicating that transitions became slower (*Figure 3F*). The higher open probability within bursts resulted from both prolonged opening (as open dwell times increased from 18.6 ± 1.2 to 28.9 ± 1.2 ms) and from an increased probability of reopening (visible as increased number of openings per burst; *Figure 3E*), consistent with an open-state stabilization that explains the reduced deactivation seen in macroscopic recordings.

The fact that the open and closed times were both affected in the overall dwell time distributions as well as in intraburst distributions suggests that α-Mangostin can bind to the open and the closed state. Accordingly, application of α-Mangostin on closed BKα channels (at –80 mV) resulted in maximal activation with the first depolarization pulse, indicating that α-Mangostin reached its site of action during the closed period (*Figure 3G*).

## Localization of the α-Mangostin binding site

We recently described the polypharmacology of the class of negatively charged activators (NCAs) in different potassium channels, specifically in TREK-1 and BK channels. The BK channel opener GoSlo-SR-5-6 (*Roy et al., 2012*; *Webb et al., 2015*) also activated TREK-1 channels, while BL-1249 known as TREK/TRAAK channel opener, also activated BK channels. We identified their common binding site in the pore close to the fenestration of TREK-1 channels and molecular dynamics (MD) simulations predicted the equivalent binding site in the pore of BK channels (*Schewe et al., 2019*). The question arose if α-Mangostin could also occupy this binding site, as in part suggested in a docking by *Kim et al., 2023*, who proposed that P183 and L304, which both are part of the NCA binding

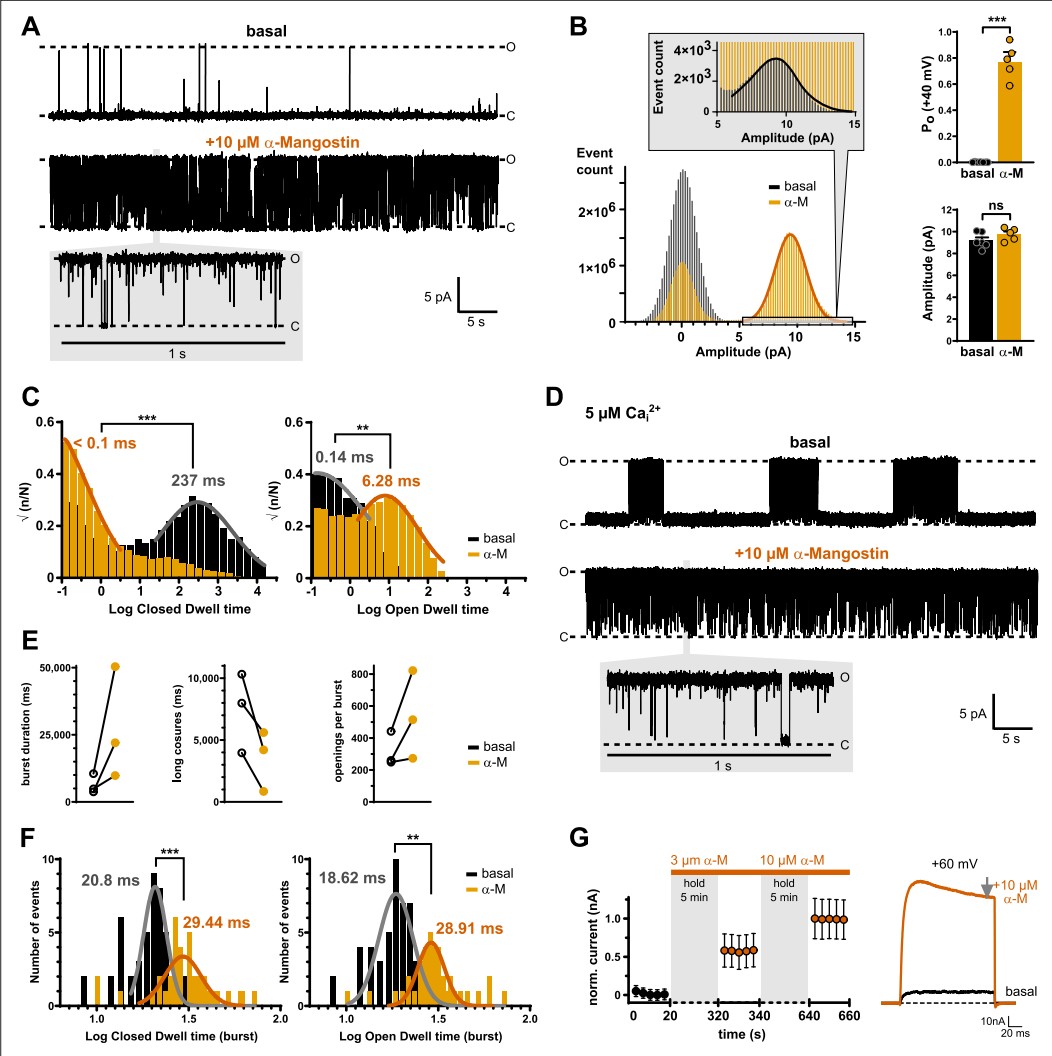

**Figure 3.** Activation mechanism of α-Mangostin. (**A**) Exemplary current traces of a single BKα channel in the basal state and after activation by 10 µM α-Mangostin (1 min recordings; inset shows 1 s; O and C denote open and closed levels). Single-channel currents were recorded at +40 mV in inside-out patches from transiently transfected HEK293 cells in symmetrical potassium gradients with 100 mM free $Ca_i^{2+}$ ($n = 4$–7). (**B**) All-points histograms with Gaussian fits for the basal and α-Mangostin-activated state, and bar graphs of the derived mean ± SEM open probabilities ($P_o$) and amplitudes. The inset magnifies the open peak in the basal state. (**C**) Closed and open dwell time histograms with fits for channels in the basal and in the α-Mangostin-activated state derived after event detection in single-channel measurements as shown in (**A**). (**D**) Exemplary current traces of a single BKα channel recorded as in (**A**), but with 5 µM free $Ca_i^{2+}$. (**E**) Burst duration, duration of long closed times, and number of openings per burst derived from burst analysis ($n = 3$ patches). (**F**) Closed and open dwell time distribution within bursts with fits for channels in the basal and in the α-Mangostin-activated state. (**G**) Normalized mean ± SEM currents of BKα channels before and after application of 3 or 10 µM α-Mangostin in the closed state. Channels were held closed at –80 mV and only very shortly pulsed to +60 mV after 5 min incubation to assess the current size/activation state of the first and the following pulses. Measurements were done in transiently transfected HEK293 cells in whole-cell mode in physiological potassium gradients with 100 nM $Ca_i^{2+}$ as shown by the representative current traces to the right, and steady-state currents were analyzed (grey arrow). Data and statistics see Source data file 1.

site, interact with α-Mangostin in TREK-1 channels. Therefore, we measured the dose-dependent α-Mangostin activation of TREK-1 wildtype channels and its 'signature' mutant of the fenestration, L304C. As expected, the apparent affinity is markedly reduced in the L304C mutant in the measurable concentration range (*Figure 4—figure supplement 1A*). Quaternary ammonium ions are known to occupy the central cavity of K+ channels as TREK-1 and BK just below the SF near the NCA binding

site (*Li and Aldrich, 2004*; *Piechotta et al., 2011*; *Fan et al., 2024*). In a competition experiment, we recorded dose–response relationships and determined the $IC_{50}$ value for tetrapentylammonium (TPA) with and without preactivation by 10 μM α-Mangostin. The TPA $IC_{50}$ value increased ≈9-fold from 0.72 ± 0.64 to 6.33 ± 0.86 μM in the presence of α-Mangostin (*Figure 4—figure supplement 1B*), showing that α-Mangostin was present in the pore and hindered the access of TPA to its binding site. Finally, cysteine scanning mutagenesis of residues in M2 and M4 revealed that mainly G181, I182, P182 on M2 and G308 in M4 were involved, while most M4 residues tested had intermediary effects (*Figure 4—figure supplement 1C*), suggesting that the binding region of α-Mangostin between M2 and M4 overlaps with the NCA binding site identified with the help of BL-1249, but is not completely identical (*Figure 4—figure supplement 1D*).

We next probed if α-Mangostin accesses the pore cavity of BKα channels, as in TREK channels, or binds elsewhere in the protein. We conducted a competition experiment as above and measured the dose-dependent inhibition by 0.1, 1, and 10 μM tetrahexylammonium (THexA; *Figure 4A*). In the presence of α-Mangostin, the inhibition by THexA was clearly reduced, and the apparent THexA affinity (as estimated $IC_{50}$) was steeply decreased ≈21-fold from 77.51 ± 5.53 nM to 1.64 ± 0.58 μM, showing that the presence of α-Mangostin hindered the access of THexA to its binding site. This is consistent with previous findings that GoSlo-RS-5-6, which binds in the BK pore, also competes with THexA (*Schewe et al., 2019*). In contrast, the presence of BC5, an activator shown to bind at the interface between the transmembrane and intracellular domains (*Zhang et al., 2022*), did not interfere with THexA binding (estimated THexA $IC_{50}$ 45.60 ± 8.05 nM; *Figure 4A*).

At least one hydroxyl group of α-Mangostin is predicted to be negatively charged in physiological pH (*National Center for Biotechnology Information, 2025*); therefore, we tested the impact of solution pH on the activation potency. Indeed, the shift in $V_{1/2}$ induced by 10 μM α-Mangostin was pH-dependent (*Figure 4C*, *Figure 4—figure supplement 2*). In pH 6, the shift was reduced to 34.71 ± 5.63 mV, while it increased to 75.97 ± 2.02 mV in pH 8.5, indicating that the negative charge may be critical for effective activation and α-Mangostin might resemble an NCA-like compound.

In addition, we used molecular docking to determine the location of a possible binding site in the BKα inner pore in the $Ca^{2+}$-bound (presumably open) and the $Ca^{2+}$-free (presumably closed) structure (*Figure 4B*, *Figure 4—figure supplement 3*). We analyzed 20 poses which were all located at the cavity wall in middle S6 between residues I308 and V319, in a pocket below the SF, without obstructing the central passageway for $K^+$ ions (*Figure 4—figure supplement 3A*). In the $Ca^{2+}$-free and $Ca^{2+}$-bound structures, three and five poses were clustered with the lowest binding energies of –8.58 and –8.64 kcal $mol^{-1}$, respectively. For the best pose in the $Ca^{2+}$-free state, possible molecular interactions were predicted for the residues I308, L312, F315, and A316, and α-Mangostin was wedged between S6 segments of adjacent helices, while in the $Ca^{2+}$-bound state where S6 moves upwards, I308 was buried and interactions shifted more to A316, resulting in a more horizontal molecule position (*Figure 4—figure supplement 3C, E*).

To functionally investigate the predicted binding site, we mutated residues in the pore-lining S6 helix with side chains facing into the cavity. Many substitutions in this region affect gating of BK channels and are involved in intersubunit S6–S6 contact (*Wu et al., 2009*; *Zhou et al., 2011*). Therefore, we chose only substitutions that were most WT-like with respect to $Ca_i^{2+}$ and voltage sensitivity, that is, I308A, L312M, A316P for the hits from the docking, and additionally S317R and Y318S as internal controls that were not predicted to be part of the binding site (*Chen et al., 2014*). We first assessed the shift in voltage activation induced by 10 μM α-Mangostin (*Figure 4D, E*). $V_{1/2}$ shifts of Y318S and S317R were not or only moderately different compared to the wildtype channel (41.46 ± 5.17 and 34.33 ± 3.37 mV). In contrast, the three mutants I308A, L312M, and in particular A316P had a markedly reduced shift in $V_{1/2}$ (19.97 ± 3.12, 27.89 ± 5.42, and 4.56 ± 1.23 mV, respectively). To ensure that the almost absent shift in A316P was not caused by a general disruption of allosteric signal transduction, we also included A316G, which showed a reduction in $V_{1/2}$ comparable to I308A and L312M (23.57 ± 2.05 mV).

Furthermore, we tested GoSlo-SR-5-6 as an alternative activator, which binds in the same pore region involving A316, and was shown to also compete with THexA (*Schewe et al., 2019*; *Zhang et al., 2022*). Like for α-Mangostin, I308A and A316P reduced the shift in $V_{1/2}$ induced by 1 μM GoSlo-SR-5-6 from 42.71 ± 3.05 mV to 25.89 ± 4.05 and 25.0 ± 2.8 mV; however, A316P did not abolish GoSlo-SR-5-6 activation, showing that the complete loss of activation for α-Mangostin was likely

Biochemistry and Chemical Biology | Structural Biology and Molecular Biophysics

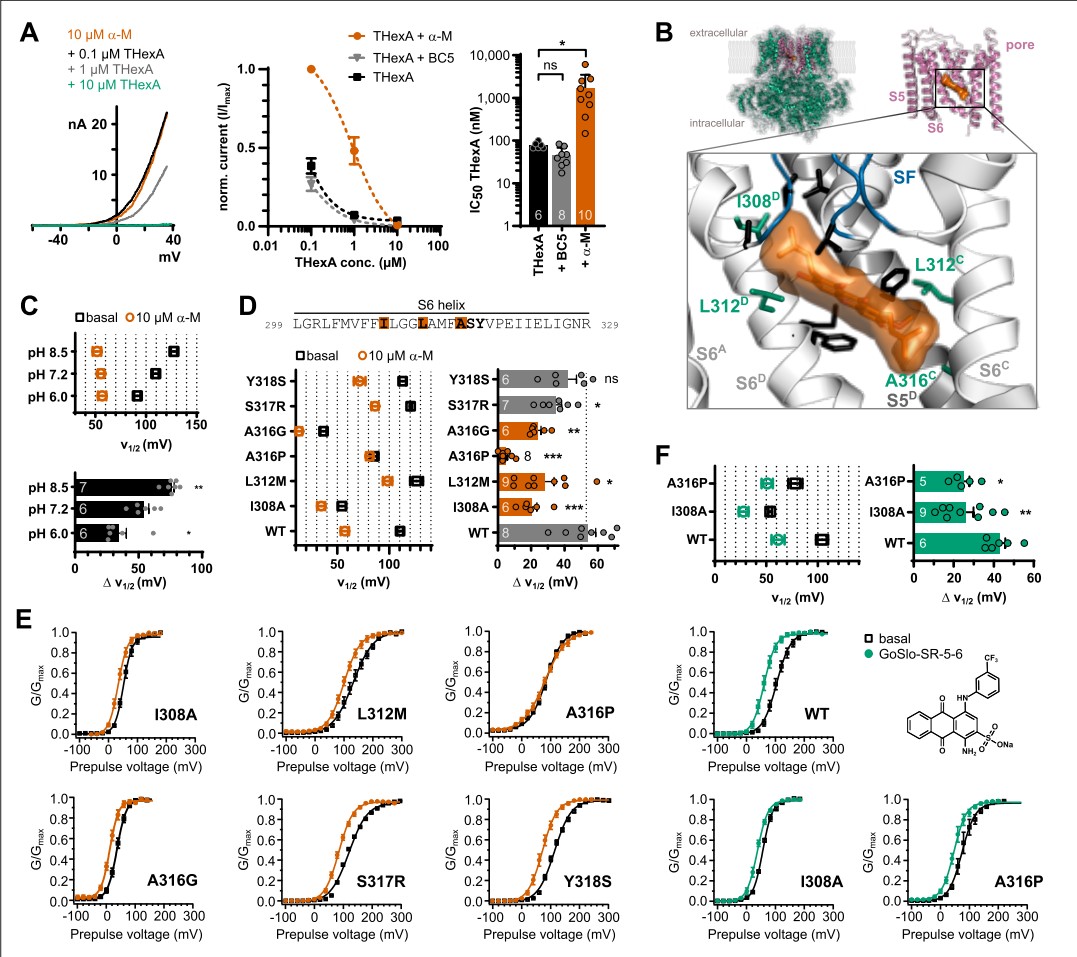

**Figure 4.** Investigation of the binding site of α-Mangostin in BKα channels. (**A**) Competition experiment showing a reduction of the block caused by THexA in the presence of α-Mangostin. Left, representative current traces for different THexA concentrations in the presence of 10 μM α-Mangostin; middle, competition experiment analysis showing the relative current of BKα channels in different THexA concentrations in the absence and in the presence of 10 μM α-Mangostin or 100 μM BC5, which does not bind in the pore; and right, estimated $IC_{50}$ values of THexA alone and in the presence of α-Mangostin or BC5. Whole-cell currents were recorded from transiently transfected HEK293 cells with a ramp protocol (−100 to +50 mV) in a physiological potassium gradient with 100 nM free $Ca_i^{2+}$ and data are shown as mean ± SEM at +40 mV. (**B**) Molecular docking of α-Mangostin to the human BK channel structure (PDB ID 6v3g, $Ca^{2+}$-free state). The full-length structure (green) was reduced to the inner pore region (pink) for the docking, and the zoom-in shows the best pose for α-Mangostin with interacting residues in stick representation; green residues mark hits from the following functional assay. Protein chain B was removed for clarity. See *Figure 4—figure supplement 3* for the $Ca^{2+}$-bound state. (**C**) Voltage of half-maximal activation ($V_{1/2}$) before and after activation by 10 μM α-Mangostin in different pH and the resulting shifts in $V_{1/2}$ ($\Delta V_{1/2}$). The pH was changed intra- and extracellularly. (**D**) Voltage of half-maximal activation ($V_{1/2}$) before and after activation by 10 μM α-Mangostin, and the resulting shifts in $V_{1/2}$ ($\Delta V_{1/2}$). (**E**) GV relationships for the six BKα mutants in the S6 segment. (**F**) Voltage of half-maximal activation ($V_{1/2}$) before and after activation by 1 μM GoSlo-SR-5-6, the resulting shifts in $V_{1/2}$ ($\Delta V_{1/2}$), and the GV relationships for the wildtype and two BKα mutants. GV relationships were measured as in *Figure 2*, and all data represent mean ± SEM. Data and statistics see Source data file 1.

The online version of this article includes the following figure supplement(s) for figure 4:

**Figure supplement 1.** Activation of TREK-1 channels by α-Mangostin and investigation of the binding region.

**Figure supplement 2.** Activation of BKα channels by 10 μM α-Mangostin in different pH.

**Figure supplement 3.** Molecular docking of α-Mangostin to the $Ca^{2+}$-free and $Ca^{2+}$-bound human BKα channel.

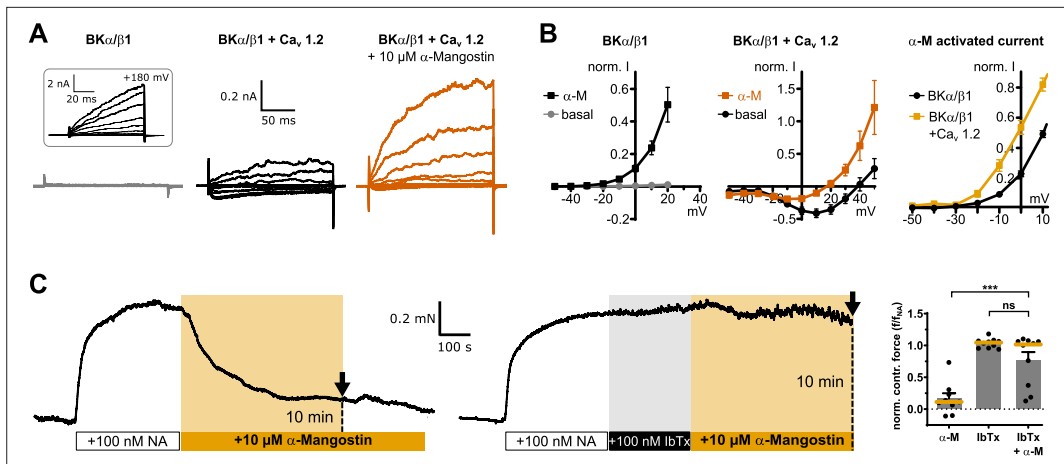

**Figure 5.** α-Mangostin activation of BK channels in physiological settings. (**A**) Representative whole-cell current traces of BKα/β1 channels alone and BKα/β1 coexpressed with Ca$_v$1.2 channels before and after application of 10 μM α-Mangostin. Currents were measured in Ca$_i^{2+}$-free conditions in a physiological potassium gradient with a family of voltage steps from –50 to +50 mV in 10 mV increments. The inset shows voltage activation with a family protocol up to +200 mV to show the presence of BKα/β1 channels. (**B**) Currents of BKα/β1 channels and BKα/β1–Ca$_v$ complexes before and after application of 10 μM α-Mangostin plotted against voltage. The last panel shows the α-Mangostin-activated currents for the range –50 to 10 mV obtained by subtracting the current before α-Mangostin application from the current after application for each potential (mean ± SEM, $n$ = 8–11 for each condition). (**C**) Representative contraction force recordings of aortic preparations from mice. 10 μM α-Mangostin were either applied directly to aortic preparations precontracted with 100 nM Noradrenaline (NA; left), or the precontracted preparations were incubated with 100 nM Iberiotoxin (IbTx) before α-Mangostin application (right). The contraction force was analyzed 10 min after α-Mangostin addition (dotted lines in recordings). The bar graph shows the normalized contraction force of preparations as mean ± SEM together with the median (orange). Data and statistics see Source data file 1. DMSO controls are shown in *Figure 5—figure supplement 1*.

The online version of this article includes the following figure supplement(s) for figure 5:

**Figure supplement 1.** Vehicle control for BKα and Ca$_v$1.2 channels and aortic tissue.

caused primarily by a loss of binding and not by an interference of the mutation with the drug transduction mechanism (*Figure 4F*). Hence, we conclude that α-Mangostin binds to and activates BKα channels via the upper S6 segment, critically involving residues I308, L312, and A316.

## α-Mangostin activation in BK–Ca$_v$ nanodomains

In resting smooth muscle cells, there is no BK channel activation within their physiological voltage range. However, in native tissue, BK channels are organized in nanodomains together with Ca$_v$ channels that can generate an increase in Ca$_i^{2+}$ concentration in the range of several orders of magnitude in their vicinity, allowing BK channels to open (*Shah et al., 2021*). This BK–Ca$_v$ complex can be restored in heterologous expression systems, and, as the intracellular solution contains 5 mM EGTA to suppress an overall increase of Ca$_i^{2+}$, BK channels will activate only when nearby Ca$_v$ channels cause a high local Ca$_i^{2+}$ concentration (*Berkefeld et al., 2006*; *Berkefeld and Fakler, 2008*). BKα/β1 channels expressed alone therefore produced virtually no current upon stepwise depolarization to +50 mV (*Figure 5A, B*), but could be activated by α-Mangostin (*Figure 5A, B*). In BKα/β1–Ca$_v$ complexes, Ca$^{2+}$ inward currents were elicited, which allowed BKα/β1 channels to open in a more negative voltage range, and these BKα/β1 currents were further enhanced by 10 μM α-Mangostin (*Figure 5A, B*). The α-Mangostin-activated current (as difference between BKα/β1-Ca$_v$ currents before and after activation) was activated at more negative potentials when BKα/β1 and Ca$_v$ channels were coexpressed than when BKα/β1 channels were expressed alone (*Figure 5B*). Hence, α-Mangostin could potentiate BKα/β1 channel currents upon local Ca$_i^{2+}$ increase through nearby Ca$_v$-channels, as it could occur upon a Ca$_i^{2+}$ spike in a cell.

## α-Mangostin relaxes vascular tissue in aortic preparations via BK channels

Cardiovascular benefits of Mangostins could arise from vasodilatory effects leading to a reduction of blood pressure. Years ago, a study reported that γ-Mangostin, in which the methoxy group of α-Mangostin is exchanged for a hydroxyl group, induced relaxation of rat aortic rings, but no molecular target was found (*Tep-Areenan and Suksamrarn, 2012*). We sought to demonstrate that Mangostin-induced vasorelaxation of native vascular tissue is indeed mediated by BK channels. Aortic preparations from mice were equilibrated at a contraction force of 2–3 mN and subsequently precontracted half-maximally with 100 nM noradrenaline. The application of 10 µM α-Mangostin quickly and efficiently induced relaxation, while relaxation was absent or greatly attenuated when the specific BK channel inhibitor Iberiotoxin (IbTx) was applied for 5–6 min before α-Mangostin was administered (*Figure 5C*). In total, α-Mangostin reduced the normalized contraction force of the preparations by more than 85% (to 0.16 ± 0.08) compared to the precontracted state, while it did not change after IbTx application alone (1.04 ± 0.002) or after preblock of BK channels (0.78 ± 0.12), proving that BK channels must mediate the vasodilative effect seen upon Mangostin treatment.

## Discussion

We provide mechanistic insight into how a natural xanthone compound modulates BK channels, which are major players in vascular function. α-Mangostin activates BK channels by shifting their voltage sensitivity and slowing deactivation, with lesser effects on activation and marked effects on $Ca_i^{2+}$ sensitivity. We show that binding of α-Mangostin is state-independent, and we provide evidence for a binding site in the gating-sensitive S6 segment.

### Mangostin binds to closed and open channels and promotes the open state

BK channel $p_o$ is controlled by $Ca_i^{2+}$, voltage, a shift of the closed–open equilibrium of the pore domain, or a combination of those stimuli (*Cui et al., 1997*). Mangostin shifted the voltage activation curve to more negative potentials, meaning an increased open probability at negative voltages. The slope of the *GV* relationship was not changed neither in homomeric BKα nor in the smooth muscle BKα/β1 channels, indicating that voltage sensitivity itself was not directly affected by α-Mangostin. Shifts in the $V_{1/2}$ values induced by α-Mangostin were present over a range of $Ca_i^{2+}$ concentrations relevant in smooth muscle cells (0.1, 1, and 10 µM), which implies that the closed–open equilibrium is shifted toward the open state by a mechanism distinct from $Ca_i^{2+}$ sensing, or that at least the $Ca_i^{2+}$ sensing mechanism is not likely to contribute significantly to α-Mangostin activation.

The activation can be mainly explained by the marked slowing of the deactivation time constants. Consistently, the mean open dwell time was increased in the single-channel measurements and brief closings within bursts were reduced, while we observed no change in amplitude with and without α-Mangostin (corresponding to a conductance of ≈240 pS). The impact on open as well as closed dwell time distributions and the instantaneous activation of macroscopic currents from the closed state suggest that the molecule binds state-independently.

In the presence of the β1 subunit, as in vascular BK channels (*Knaus et al., 1994*), the activation characteristics were enhanced. The activation time course was little affected in BKα channels, but in BKα/β1 channels an acceleration of the activation caused by α-Mangostin became more prominent, counteracting the slow activation kinetics brought by the β1 subunit (*McManus et al., 1995*). Additionally, the slow deactivation time course in the presence of the β1 subunit was further slowed down remarkably. As the affinity (as estimated $EC_{50}$) was very similar and the binding site is located in the S6 segment in the inner pore, the stronger shift in the $V_{1/2}$ value is likely not due to changes in binding, but reflects the enhanced shift of the closed–open equilibrium toward the open state conferred by the β1 subunit. Such β-subunit-dependent effects have also been reported for other modulators, for example, DHS-I (*McManus et al., 1993*; *Giangiacomo et al., 1998*), DHA (*Hoshi et al., 2013a*), arachidonic acid (*Martín et al., 2021*), 17β-estradiol (*Valverde et al., 1999*), and some GoSlo compounds (*Large et al., 2015*).

## The binding site of α-Mangostin includes residues in the gating-sensitive S6 segment

We demonstrated the presence of α-Mangostin in the channel pore by competition for binding with quaternary ammonium ions known to bind below the SF (*Li and Aldrich, 2004*; *Fan et al., 2024*), and predicted by molecular docking to the $Ca^{2+}$-free and the $Ca^{2+}$-bound state, identified residues in the S6 segment that are part of the binding site.

α-Mangostin binds in the cleft formed by the S6 helices of adjacent subunits, and substitution of the nonpolar residues I308, L312, and especially A316 most strongly reduced the shift in $V_{1/2}$ induced by α-Mangostin. All residues are known to participate in gating: they are part of a hydrophobic network within S6 that acts as a hinge upon channel opening, and polar substitutions strongly favor the open state (*Zhou et al., 2011*; *Chen et al., 2014*; *Hite et al., 2017*). L312 has further been shown to stabilize intersubunit interaction with F315, whose destruction opens the channel (*Wu et al., 2009*). In the open state, S6 is thought to move upwards, as seen in the $Ca^{2+}$-bound cryo-EM structure (*Kallure et al., 2023*; *Yamanouchi et al., 2023*; *Redhardt et al., 2024*), thereby burying the I308 sidechain and diminishing the dimension of the upper cleft. Therefore, the contribution of I308 to α-Mangostin activation is likely allosteric in nature, and the pose of the molecule changes with the S6 position.

The α-Mangostin binding region overlaps with the reported binding site of the NCA GoSlo-SR-5-6, where L312, A316, S317, and V319 are involved (*Webb et al., 2015*; *Schewe et al., 2019*), but it is not identical. When we functionally investigated the α-Mangostin binding site close to the fenestration in TREK-1 channels, we also found only a partial overlap with key residues involved in the NCA binding site. Interaction with P183 and L304 was consistent with an earlier molecular docking (*Kim et al., 2023*), but the strongest reduction of activation was found for residues in M2 rather than M4.

## Mechanism of mangostin activation

We expect that small structural or electrostatic changes in this gating-sensitive region strongly affect the closed–open equilibrium (*Wu et al., 2009*; *Zhou et al., 2011*; *Chen et al., 2014*). Previous studies suggest several alternative mechanisms by which α-Mangostin could induce channel opening. We have shown that NCAs bind to a region below the SF in several $K^+$ channels, such as $K_{2P}$ (e.g., TREK), hERG, or BK channels to induce channel opening (*Schewe et al., 2019*). We speculated that the negative charge common to these activators might alter SF ion occupancy via an electrostatic mechanism, thereby promoting SF opening. Our finding that the potency of α-Mangostin depends on pH suggests that a negative charge may also be critical for its activation mechanism. However, the importance of the SF for BK gating remains unresolved. Furthermore, α-Mangostin also did not increase the single-channel conductance as seen for NCAs in TREK-2 channels, which are known to be gated at the SF (*Schewe et al., 2019*). Therefore, the contribution of the SF to the Mangostin effect requires further investigation. Alternatively, α-Mangostin binding to the gating-critical S6 region could induce gating at a putative lower gate; however, such a gate is currently speculative, as the available BK structures lack clear evidence for a lower gate (but the final closed state may still be elusive) (*Hite et al., 2017*; *Tao et al., 2017*).

In addition to the concepts of an SF gate or a lower gate, hydrophobic gating has been proposed as an alternative gating mechanism in BK channels. MD simulations suggest that the pore of metal-free (presumably closed) BK structures undergoes dewetting, whereas the pore of metal-bound (presumably open) structures remains hydrated (*Jia et al., 2018*; *Gu and de Groot, 2023*). Thus, binding of negatively charged α-Mangostin might prevent or antagonize dewetting of the hydrophobic cavity and thereby stabilize the conductive pore state, probably with substantial involvement of I308. Indeed, a recent MD study suggested that the NCA NS11021, a smooth muscle relaxant with a biarylthiourea structure, can enter the dewetted BK pore to promote hydration (*Rockman et al., 2020*; *Nordquist et al., 2024*). However, the same MD simulations also indicated that NS11021 has no stable binding pose but engages in various hydrophobic interactions with pore-cavity residues. In contrast, our results suggest a more stable binding pose with specific interactions to a number of residues in the $Ca^{2+}$-free as well as in the $Ca^{2+}$-bound state, as inferred from our mutagenesis and docking data. Clearly, further work is required to elucidate the mechanism of BK channel gating and the exact way Mangostin and other NCAs modulate this process to promote channel activation.

## Activation of BK channels in nanodomains and relaxation of aortic tissue

Our data demonstrate that α-Mangostin is able to potentiate $Ca_v$-induced currents, and that α-Mangostin-activated BK channels are responsible for the relaxation of mouse vascular tissue in a conceptual ex vivo experiment, providing a very probable explanation for the reduction of blood pressure reported in a recent animal study (*Xu et al., 2024*).

We reconstructed nanodomains of BK and $Ca_v1.2$ channels, as for example present in arterioles (*Berkefeld et al., 2006*; *Tykocki et al., 2017*). In smooth muscle cells, BK activation is strictly coupled to changes in $Ca_i^{2+}$. Physiological $Ca_i^{2+}$ concentrations lie between 100 nM in the resting state, 0.4–1 μM upon activation of a $Ca^{2+}$ mobilizing receptor (*Savineau and Marthan, 2000*; *Hill-Eubanks et al., 2011*), and up to several ten micromolar (10–40 μM) when sparks form locally (*Zhuge et al., 2002*; *Berkefeld et al., 2010*). Transient outward currents through activated BK channels then act as negative feedback, repolarize the cell, and terminate $Ca^{2+}$ influx (*Eisvand et al., 2022*). We demonstrated that the α-Mangostin-activated current is potentiated by the coupling to $Ca_v$ channels. BKα/β1 currents in the physiological voltage range of smooth muscle cells (−50 to –30 mV) would be very small in resting $Ca_i^{2+}$, while the $p_o$ would be already very high when exceeding 10 μM $Ca^{2+}$, leaving less room for activation. α-Mangostin would particularly impact potentiation of BK currents upon physiological $Ca^{2+}$ elevation (i.e., between 1 and 10 μM) as well as in the physiological voltage range, and would synergistically lower the BKα/β1 activation threshold to enhance the negative feedback mechanism.

Accordingly, our investigation of precontracted aortic tissue from mice showed a remarkable and robust relaxation after application of α-Mangostin. Such relaxation of aortic rings after application of α- or γ-Mangostin was reported before (*Chairungsrilerd et al., 1996*; *Chairungsrilerd et al., 1997*; *Tep-Areenan and Suksamrarn, 2012*), but the exact molecular target remained inconsistent. We did not assess the effects of Mangostin on other components of the contraction pathway, such as Cav channels or ryanodine receptors, and therefore cannot exclude their modulation. However, given the robust activation of BK channels and the complete loss of relaxation following preincubation with the selective BK inhibitor iberiotoxin, our data indicate that BK channels represent the primary molecular target of Mangostins in vascular smooth muscle cells.

Interestingly, relaxation was also reported for other xanthone derivatives (*Cheng and Kang, 1997*; *Wang et al., 2002*; *Capettini et al., 2009*; *Câmara et al., 2010*; *Diniz et al., 2013*), which legitimizes further research of substituted xanthones as pharmacophore. Robust data on bioavailability and the existence and nature of active metabolites are currently lacking. A small study claimed that α-Mangostin was bioavailable in humans after supplement ingestion (*Kondo et al., 2009*). Animal research analyzed metabolites and showed that maximal plasma concentrations are reached 1 hr after oral intake in mice (*Ramaiya et al., 2012*; *Petiwala et al., 2014*; *Han et al., 2015*). However, no metabolites have been functionally investigated to help in optimization of the Mangostin pharmacophore; however, our finding that γ-Mangostin has a slightly higher potency in activating TREK-1 and BKα channels may be a starting point.

Our screen across different $K^+$ channel families revealed that other representatives of the $K_{2P}$ (TWIK-1 and TASK-3) were particularly inhibited by more than 60%. This could also link other claimed benefits of Mangostin to their molecular targets. Importantly, hERG currents were unaffected, mitigating the cardiotoxicity risk upon consumption of Mangostin nutraceuticals. BK channel function is decreased in chronic conditions that are often associated with the lack of a balanced diet and age, such as metabolic syndrome, diabetes, and obesity (*Tykocki et al., 2017*). Consequences such as increased vascular tone and hypertension, ultimately leading to reduced cardiovascular health and premature death, may therefore be addressed with Mangostin xanthones as a potential new class of BK channel activators.

## Materials and methods

**Key resources table**

| Reagent type (species) or resource | Designation | Source or reference | Identifiers | Additional information |
|---|---|---|---|---|
| Gene (*Homo sapiens*) | *KCNK2* | GenBank | NM_001017425.2 | hTREK-1b |
| Gene (*Homo sapiens*) | *KCNK16* | GenBank | NM_032115.3 | hTALK-1 |
| Gene (*Homo sapiens*) | *KCNK18* | GenBank | NM_181840.1 | hTRESK |
| Gene (*Homo sapiens*) | *KCNK1* | GenBank; **Feliciangeli et al., 2010** | NM_002245.3 | hTWIK-1$^m$; I293A, I294A |
| Gene (*Homo sapiens*) | *KCNK6* | GenBank; **Bobak et al., 2017** | NM_004823.1 | hTWIK-2$^m$; I289A, L290A |
| Gene (*Homo sapiens*) | *KCNK9* | GenBank | NM_001282534.1 | hTASK-3 |
| Gene (*Homo sapiens*) | *KCNK13* | GenBank | NM_022054.3 | hTHIK-1 |
| Gene (*Mus musculus*) | Kcnma1 | GenBank | NM_001014797.3 | mBKα (Slo1.1); G/S-rich N-terminus removed, corresponding to residues 65–1236 of the native channel |
| Gene (*Mus musculus*) | Kcnmb1 | GenBank | NM_031169.4 | mβ1 |
| Gene (*Homo sapiens*) | *KCNA1* | GenBank | NM_000217.3 | hK$_v$ 1.1 |
| Gene (*Homo sapiens*) | *KCNA3* | GenBank | NM_002232.5 | hK$_v$ 1.3 |
| Gene (*Homo sapiens*) | *KCNH2* | GenBank | AJ512214.1 | hK$_v$ 11.1 (hERG1b) |
| Gene (*Homo sapiens*) | *KCNJ1* | GenBank | NM_000220.4 | hK$_{ir}$ 1.1 (ROMK) |
| Gene (*Mus musculus*) | KCNJ2 | GenBank | NM_008425.4 | mK$_{ir}$ 2.1 |
| Gene (*Rattus norvegicus*) | Cacna1c | GenBank | M67515.1 | Ca$_v$α 1C |
| Gene (*Rattus norvegicus*) | Cacnb1 | GenBank | NM_017346.1 | Ca$_v$ β1 |
| Gene (*Rattus norvegicus*) | Cacna2d1 | GenBank | AF286488 | Ca$_v$ α2δ1 |
| Gene (synthetic) | *EYFP* | Addgene | https://www.addgene.org/vector-database/2688/ | pEYFP |
| Strain, strain background (*Escherichia coli*) | DH5α | NEB | Cat. #: C2987 | Chemically competent cells |
| Strain, strain background (*Mus musculus*) | CD1/CHR2 Mice (1 female, 7 male) | IMSR | RRID:IMSR_CRL:022 | age 5–6 months |
| Cell line (*Homo sapiens*) | HEK293 | Sigma-Aldrich/ Cellosaurus | RRID:CVCL_0045 | |
| Cell line (*Cricetulus griseus*) | CHO-K1 | Sigma-Aldrich/ Cellosaurus | RRID:CVCL_0214 | |
| Peptide, recombinant protein | Iberiotoxin | Alomone Labs (Jerusalem, Israel) | Cat. #: STI-400 | |
| Commercial assay or kit | Lipofectamine 2000 | Invitrogen, Thermo Fisher, Schwerte, Germany | Cat. #: 11668019 | |

| Reagent type (species) or resource | Designation | Source or reference | Identifiers | Additional information |
|---|---|---|---|---|
| Commercial assay or kit | FuGENE | Promega, Walldorf, Germany | Cat. #: E2311 | |
| Commercial assay or kit | Venor GEM OneStep | Minerva Biolabs | Cat. #: 11-8025 | Mycoplasm PCR test |
| Chemical compound, drug | α-Mangostin | Merck (Darmstadt, Germany) | Cat. #: M3824 | |
| Chemical compound, drug | γ-Mangostin | Merck (Darmstadt, Germany) | Cat. #: M6824 | |
| Chemical compound, drug | Mangosteen antioxidant support | https://Swanson.com | Cat. #: SWH259 | |
| Chemical compound, drug | BC-5 | MP Biomedicals (Irvine, USA) | Arg-4-methoxy-2-naphthylamine | |
| Chemical compound, drug | TPA | Merck (Darmstadt, Germany) | Cat. #: 241970 | |
| Chemical compound, drug | THexA | Merck (Darmstadt, Germany) | Cat. #: 263834 | |
| Chemical compound, drug | GoSlo-SR-5-6 | Courtesy of Mark Hollywood (Dundalk Institute of Technology) | Sodium 1-amino-4-((3trifluoromethylphenyl) amino)-9,10-dioxo-9,10-dihydroanthracene-2-sulfonate | |
| Software, algorithm | Patchmaster v2.78 | HEKA Elektronik (Lambrecht, Germany) | RRID:SCR_000034 | |
| Software, algorithm | Fitmaster v2.92 | HEKA Elektronik (Lambrecht, Germany) | RRID:SCR_016233 | |
| Software, algorithm | PyMOL | *Schrodinger LLC, 2015* | RRID:SCR_000305 | |
| Software, algorithm | AutoDock4 v4.2.6/ MGL Tools v1.5.7 | *Morris et al., 2009* | RRID:SCR_012746 | |
| Software, algorithm | LabChart | AD Instruments, Mannheim, Germany | RRID:SCR_018833 | |
| Software, algorithm | IgorPro | WaveMetrics, Portland, USA | RRID:SCR_000325 | |
| Software, algorithm | pClamp Clampfit v11.2.2.17 | Molecular Devices, San Jose, USA | RRID:SCR_011323 | |
| Other | EPC10 | HEKA Elektronik (Lambrecht, Germany) | RRID:SCR_018399 | |

## Substances

α-Mangostin, γ-Mangostin, TPA, THexA, MgATP, NaGTP, and noradrenaline were purchased from Merck (Darmstadt, Germany). BC5 (Arg-4-methoxy-2-naphthylamine) was obtained from MP Biomedicals (Irvine, USA), and Iberiotoxin from Alomone Labs (Jerusalem, Israel). GoSlo-SR-5-6 (sodium 1-amino-4-((3trifluoromethylphenyl)amino)-9,10-dioxo-9,10-dihydroanthracene-2-sulfonate) was provided by Mark Hollywood. Mangostane food supplement stating a content of 50 mg α-Mangostin per capsule was ordered online (https://Swanson.com). Aiming for a 10 mM stock solution with respect to α-Mangostin, 205 mg of the powder found in the capsules was dispersed in 5 ml DMSO, brought into solution by vortexing and ultrasonication, and filtered through a 0.45 μM PTFE membrane to remove remaining particulate matter. TPA, THexA, noradrenaline, BC5, and Iberiotoxin were prepared as 1:1000 stocks in $H_2O$, and the Mangostins and GoSlo-RS-5-6 in DMSO at 50 mM. Aliquots were stored at –20°C and diluted to the final concentration in extracellular solution. The final DMSO concentration did not exceed 0.025%.

## Molecular biology, cell culture, and transfection

Coding sequences for channels and subunits listed in *Table 1* were subcloned in pcDNA3.1 (https://www.addgene.org/vector-database/2097/) or pFAW vectors containing a CMV promoter for

**Table 1.** Channels and subunits used in this study.

| Name | Organism | Gene name | Genbank Acc. No. |
|---|---|---|---|
| hTREK-1b | *Homo sapiens* | *KCNK2* | NM_001017425.2 |
| hTALK-1 | *Homo sapiens* | *KCNK16* | NM_032115.3 |
| hTRESK | *Homo sapiens* | *KCNK18* | NM_181840.1 |
| hTWIK-1$^m$ | *Homo sapiens* | *KCNK1* | NM_002245.3 |
| hTWIK-2$^m$ | *Homo sapiens* | *KCNK6* | NM_004823.1 |
| hTASK-3 | *Homo sapiens* | *KCNK9* | NM_001282534.1 |
| hTHIK-1 | *Homo sapiens* | *KCNK13* | NM_022054.3 |
| mBKα (Slo1.1)* | *Mus musculus* | *Kcnma1* | NM_001014797.3 |
| mβ1 | *Mus musculus* | *Kcnmb1* | NM_031169.4 |
| h$K_v$ 1.1 | *Homo sapiens* | *KCNA1* | NM_000217.3 |
| h$K_v$ 1.3 | *Homo sapiens* | *KCNA3* | NM_002232.5 |
| h$K_v$ 11.1 (hERG1b) | *Homo sapiens* | *KCNH2* | AJ512214.1 |
| h$K_{ir}$ 1.1 (ROMK) | *Homo sapiens* | *KCNJ1* | NM_000220.4 |
| m$K_{ir}$ 2.1 | *Mus musculus* | *KCNJ2* | NM_008425.4 |
| $Ca_v$ 1.2: | | | |
| $Ca_v$α 1C | *Rattus norvegicus* | *Cacna1c* | M67515.1 |
| $Ca_v$ β1 | *Rattus norvegicus* | *Cacnb1* | NM_017346.1 |
| $Ca_v$ α2δ1 | *Rattus norvegicus* | *Cacna2d1* | AF286488 |

*G/S-rich N-terminus removed, corresponding to residues 65–1236 of the native channel.

expression in cultured cells, or in the pBF *Xenopus* oocyte vector for low expression in the single-channel experiments (see below). Amino acid substitutions were introduced by site-directed mutagenesis PCR according to the QuikChange protocol (Stratagene, La Jolla, USA) with custom primers containing the desired base exchange. All constructs were verified by Sanger sequencing. To enhance plasma membrane expression, the retrieval motif was removed in hTWIK-1 and hTWIK-2 channels by the substitutions I293A, I294A (*Feliciangeli et al., 2010*) and I289A, L290A (*Bobak et al., 2017*), respectively, yielding hTWIK-1$^m$ and hTWIK-2$^m$ channels.

HEK293 cells (RRID:CVCL_0045, Sigma-Aldrich) were cultivated in Dulbecco's modified Eagle's medium supplemented with 10% FCS and penicillin/streptomycin (100 U ml$^{-1}$/100 µg ml$^{-1}$) in 5% $CO_2$ at 37°C. For cultivation of CHO-K1 cells (RRID:CVCL_0214, Sigma-Aldrich), the medium was additionally supplemented with non-essential amino acid mix (Gibco MEM-NEAA; Thermo Fisher, Schwerte, Germany) and 10 mM HEPES. Cell lines were regularly tested for mycoplasm contamination by PCR (Venor GEM OneStep, Minerva Biolabs) and cell identity was visually controlled by comparison to the ATCC database.

HEK293 cells were transiently transfected with Lipofectamine 2000 (Invitrogen, Thermo Fisher, Schwerte, Germany) or FuGENE (Promega, Walldorf, Germany) according to the manufacturer's instructions and incubated overnight. pEYFP (https://www.addgene.org/vector-database/2688/) was included as a transfection marker. In BKα/β1 coexpressions, the β1 subunit was used in 10-fold excess to ensure a uniform channel population. Cells were trypsinized and seeded onto glass coverslips 2–3 hr prior to electrophysiological experiments.

## Electrophysiology

Voltage clamp experiments were conducted with a HEKA EPC10 amplifier (RRID:SCR_018399) controlled by Patchmaster Software (v2.78; RRID:SCR_000034; HEKA Elektronik, Lambrecht, Germany). Standard measurements were done in transiently transfected HEK293 cells in the whole-cell

configuration in physiological potassium gradients. Pipettes were pulled from thin-walled borosilicate glass, fire-polished, and had resistances of 1.4–2 MΩ. Series resistance was compensated to at least 70%. The extracellular solution contained (in mM): 135 NaCl, 5 KCl, 2 CaCl$_2$, 2 MgCl$_2$, 10 D(+)-glucose, 10 HEPES (pH 7.3). The intracellular solution was (in mM): 140 KCl, 2 MgCl$_2$, 1 CaCl$_2$, 2.5 EGTA, 10 HEPES (pH 7.3), corresponding to ≈100 nM free Ca$^{2+}$. For K$_{ir}$ 1.1 and 2.1 channel recordings, 3 mM MgATP and 0.3 mM NaGTP were included. Where indicated, K$^+$ was substituted for Cs$^+$ to reduce the outward currents of BK channels. Cells were held at –80 mV and a voltage ramp from –100 to +60 mV of 1 s duration was applied every 5 s. For K$_v$ channels, a rectangle pulse was used as indicated in the figure legends.

The activation of BK channels was measured as conductance $G$ in symmetrical potassium solutions to ensure a unitary conductance across the voltage range. Cs$^+$ was used as the intracellular ion to reduce outward currents. The extracellular solution contained (in mM): 140 KCl, 2 CaCl$_2$, 2 MgCl$_2$, 10 D(+)-glucose, 10 HEPES (pH 7.3). The intracellular solution was (in mM): 140 CsCl, 2 MgCl$_2$, 1 CaCl$_2$, 2.5 EGTA, 10 HEPES (pH 7.3), corresponding to ≈100 nM free Ca$_i^{2+}$ (calculated with WEBMAXC). For recordings with 1 µM and 10 µM free Ca$_i^{2+}$, CaCl$_2$ and EGTA were raised to 2/2.2 and 2/4.04 mM, respectively. Cells were held at negative holding potentials and currents were elicited with a family of rectangle voltage pulses in 20 mV increments to positive potentials as needed for saturation of the tail currents, followed by a repolarization step to elicit the tail currents as detailed in the figure legends. The currents were analyzed with Fitmaster (HEKA Elektronik, Lambrecht, Germany; RRID:SCR_016233), and conductance–voltage ($GV$) relationships were generated from normalized tail currents and fit to a standard Boltzmann relationship to obtain $V_{1/2}$ values. Individual fits were then averaged for mean $V_{1/2}$ values.

$$\frac{G}{G_{max}} = \frac{1}{1 + e^{\frac{(V - V_{1/2})}{k}}}$$

where $G_{max}$ is the maximal tail current, $V_{1/2}$ is the voltage of half-maximal activation of the current, and $k$ is the slope factor.

Activation in different pH was measured using the following solutions: Extracellular solution pH 8.5 contained (in mM): 140 KCl, 2 CaCl$_2$, 2 MgCl$_2$, 10 D(+)-glucose, 10 Ampso; intracellular solution pH 8.5 contained (in mM): 140 CsCl$_2$, 10 Ampso, 2 MgCl$_2$, 1 CaCl$_2$, 2.05 EGTA. Extracellular solution pH 6 was (in mM): 140 KCl, 2 CaCl$_2$, 2 MgCl$_2$, 10 D(+)-glucose, 10 MES; intracellular solution pH 6 contained (in mM): 140 CsCl$_2$, 10 MES, 2 MgCl$_2$, 0.5 CaCl$_2$, 11 EDTA. The pH was changed intra- and extracellularly.

In TPA competition experiments for TREK-1 channels, a dose–response relationship in the absence and presence of α-Mangostin was recorded and the IC$_{50}$ was obtained from a Hill fit. The THexA competition experiments for BKα channels were conducted as 3-point determinations using 0.1 and 1 µM THexA as concentrations close to half-maximal inhibition without and with α-Mangostin and 10 µM THexA for full block. To estimate the IC$_{50}$, a simplified 4-parameter logistic regression was used, assuming $a=0$ and $b=1$.

$$y = d + \frac{a-d}{1 + \left(\frac{x}{c}\right)^b} \rightarrow c = \frac{x}{\left(\frac{d}{y} - 1\right)}$$

where $a$: lower asymptote, $b$: slope, $c$: IC$_{50}$, $d$: upper asymptote, $x$: THexA concentration, and $y$: current.

## Single-channel recordings

Single channels were recorded at room temperature from excised inside-out patches of HEK293 cells transiently transfected with BKα in a continuous voltage protocol at +40 mV. To increase the chances of obtaining patches with only one channel, a pBF vector with a suboptimal beta globin promoter yielding only low expression was used. Pipette resistances were 12–15 MΩ, and symmetrical intra- and extracellular solution contained (in mM) 140 KCl, 2 MgCl$_2$, 1 CaCl$_2$, 2.5 EGTA, 10 HEPES (pH 7.3 with KOH), corresponding to ≈100 nM free Ca$^{2+}$, or 140 KCl, 2 MgCl$_2$, 2.36 CaCl$_2$, 2.4 EGTA, 10 HEPES (pH 7.3 with KOH), corresponding to ≈5 µM free Ca$^{2+}$. Currents were recorded at a sampling rate of 100 kHz with a final bandwidth $f_c$ of 7.4 kHz using an EPC10

amplifier (RRID:SCR_018399) controlled by Patchmaster software (v2.78; RRID:SCR_000034; HEKA Elektronik, Lambrecht, Germany; RRID:SCR_000034). Traces were analyzed in Clampfit (v11.2.2.17; RRID:SCR_011323; Molecular Devices, San Jose, USA). The filter rise time $T_r = \frac{0.3321}{f_c}$ was 45 µs, and openings $>2$ $T_r$ were fitted (*Colquhoun and Sigworth, 1995*). Histograms were calculated in Clampfit and fitted with a Gaussian function to obtain amplitudes and dwell times using Prism (v8.4.3; GraphPad Software Inc, San Diego, USA; RRID:SCR_002798). Single-channel conductance was calculated from $g = \frac{I}{(V_m - E_{K^+})}$, where $g$, conductance; $I$, single-channel current; $V_m$, membrane voltage; $E_{K^+}$, K$^+$ equilibrium potential. Bursts were detected using a minimum closed interval of 200 ms. Bursts with a single opening (0 ms intraburst interval) were included in the calculation of mean closed times within a burst. Means were calculated per patch. The fits of kinetic distribution data are descriptive as data was pooled.

## Reconstitution of nanodomains

BKα/β1–Ca$_v$ complexes were restored as described by *Berkefeld et al., 2006* with the following modification: For nanodomain formation of Ca$_v$ 1.2 and BKα/β1, CHO-K1 cells were seeded onto coverslips and microinjected (InjectMan 4, Eppendorf, Hamburg, Germany) with a DNA mixture of Ca$_v$α 1C, Ca$_v$ β1, Ca$_v$ α2δ1, BKα, β1, and EYFP in the ratio 10:10:10:10:100:1 to ensure the presence of all subunits. Cells were incubated overnight, and currents were recorded from EYFP-positive cells 14–18 hr after injection. Cells were measured in a physiological potassium gradient with a voltage step protocol from –50 to +60 mV in 10 mV increments. The extracellular solution contained (in mM): 135 NaCl, 5 KCl, 2 CaCl$_2$, 2 MgCl$_2$, 10 D(+)-glucose, 10 HEPES (pH 7.3). The intracellular solution was (in mM): 140 KCl, 2 MgCl$_2$, 5 EGTA, 10 HEPES (pH 7.3). α-Mangostin was added to the bath solution at a final concentration of 10 µM.

## Molecular docking

The input structures for molecular docking were generated by truncating the structure of the human BKα channel in the Ca$^{2+}$-free and Ca$^{2+}$-bound state (PDB ID 6v3g, 6v38) to only the pore region (res T229-R329) using PyMOL (*Schrodinger LLC, 2015*; RRID:SCR_000305). Docking was performed with AutoDock4 (v4.2.6; RRID:SCR_012746) using MGL Tools (v1.5.7) (*Morris et al., 2009*). A run with lower resolution grid was done, and after α-Mangostin was seen in the pore, the docking was repeated with a gridbox restricted to the cavity below the SF. Clustering and interactions were analyzed with the built-in functions of MGL Tools.

## Aortic smooth muscle preparations

Animal care and use and the procedures to obtain aortic tissue from mice were licensed and performed according to German animal protection law, the regulations of the state authorities of Mecklenburg-West Pomerania, and the institutional standards of the University of Rostock (Tierschutz-G_Nr. A414-18). 1 female and 7 male CD1/CHR2 mice (RRID:IMSR_CRL:022) at the age of 5–6 months were randomly chosen and killed by decapitation following isoflurane-narcosis, and the aorta was removed. Inclusion/exclusion criteria did not apply, and all animals were used for further study. The aortic wall was dissected, and tissue strips of ≈15 × 2 mm were obtained using the spiral cut technique (*Peiper and Schmidt, 1972*). Aortic preparations were tethered to glass holders and transferred to temperature-controlled 37°C bath chambers filled with Krebs solution (pH 7.4; in mM: 112 NaCl, 4.7 KCl, 2.5 CaCl$_2$, 1.2 MgCl$_2$, 1.2 KH$_2$PO$_4$, 25 NaHCO$_3$, and 11.5 glucose). The bath solution was bubbled with 95% O$_2$/5% CO$_2$. The contraction force was recorded with mechanoelectric transducers (Fort100, WPI, Sarasota, USA), connected to a PowerLab controlled by LabChart (AD Instruments, Mannheim, Germany; RRID:SCR_018833). A prestrain of 2.5–3.9 mN was applied, depending on the thickness of preparations, and strips equilibrated for ca. 45 min. The NA concentration necessary to achieve half-maximal contraction was determined by measuring a dose–response relationship, and 100 nM NA were subsequently used for precontraction. After a plateau was reached (2–4 min), α-Mangostin and IbTx were diluted into the bath solution to their final concentrations of 10 µM (final DMSO concentration 0.1%) and 100 nM, respectively, and the contraction force was recorded for ca. 20 min. Recordings were analyzed with LabChart.

## Data analysis and statistics

All data are given as mean ± SEM and the number of data points (cells or patches) is given above the bars or in the figure legends. Mean data underlying figures is reported in Source data file 1. Measurements were made from at least two different transfections. Mean currents were analyzed with Patchmaster (HEKA Elektronik, Lambrecht, Germany; RRID:SCR_000034). Fitmaster (HEKA Elektronik, Lambrecht, Germany; RRID:SCR_016233) was used to generate *GV* relationships and fits with the Boltzmann function, and for the monoexponential fits to obtain activation and deactivation time constants. $EC_{50}/IC_{50}$ values were obtained by fitting a Hill equation to the dose–response data using IgorPro (v6.3.7; WaveMetrics, Portland, USA; RRID:SCR_000325) or Prism (v8.4.3; GraphPad Software Inc, San Diego, USA; RRID:SCR_002798).

$$I_0 + \frac{I_{max} - I_0}{1 + \left(\frac{c_{1/2}}{c}\right)^h}$$

with $I_0$, basal current; $I_{max}$, activated current; $c$, concentration; $c_{1/2}$, $EC_{50}/IC_{50}$; $h$, Hill coefficient.

Statistical analysis was conducted with the built-in functions of Prism. If data were not normally distributed, nonparametric tests were used. Variances were compared with *F*-tests. Two means were then compared with a paired or unpaired two-tailed *t*-test. Three or more groups were compared using one-way ANOVA when variances and standard deviation were considered equal by Bartlett's test; otherwise, the Brown–Forsythe and Welch ANOVA was applied. Dunnett's T3 post hoc test was used to account for multiple comparisons in ANOVA, and Dunn's post hoc test was used for nonparametric tests. Statistical tests and p values are reported in Source data file 1. Symbols for p values are used in the figures as follows: ***$p < 0.001$, **$p \leq 0.01$, *$p \leq 0.05$, ns: $p \geq 0.05$.

Structures were visualized with PyMOL (*Schrodinger LLC, 2015*); RRID:SCR_000305 and figures were assembled with Inkscape v1.3 (GNU General Public License, v3; RRID:SCR_014479).

## Acknowledgements

Plasmids containing the BK β1 subunit and the different Ca$_v$1.2 subunits were a gift of Bernd Fakler (Universität Freiburg). K$_v$1.3 was kindly provided by Heinrich Terlau (Christian-Albrechts-Universität zu Kiel). GoSlo-SR-5-6 was received by courtesy of Mark Hollywood (Dundalk Institute of Technology). We thank Michaela Unmack, Sandra Grüssel, Henning Janssen, and Petra Breiden for excellent technical assistance.

## Additional information

### Funding

| Funder | Grant reference number | Author |
| --- | --- | --- |
| Deutsche Forschungsgemeinschaft | 506373940 | Thomas Baukrowitz Marianne A Musinszki |

The funders had no role in study design, data collection, and interpretation, or the decision to submit the work for publication.

### Author contributions

Soenke Cordeiro, Formal analysis, Investigation, Writing – review and editing; Robert Patejdl, Resources, Investigation, Methodology; Thomas Baukrowitz, Resources, Funding acquisition, Writing – review and editing; Marianne A Musinszki, Conceptualization, Formal analysis, Funding acquisition, Investigation, Visualization, Writing – original draft, Writing – review and editing

### Author ORCIDs

Soenke Cordeiro https://orcid.org/0000-0002-1049-8303
Robert Patejdl https://orcid.org/0000-0003-4587-4054
Thomas Baukrowitz https://orcid.org/0000-0003-4562-0505
Marianne A Musinszki https://orcid.org/0000-0002-5597-6735

### Ethics

Animal care and use and the procedures to obtain aortic tissue from mice were licensed and performed according to German animal protection law, the regulations of the state authorities of Mecklenburg-West Pomerania, and the institutional standards of the University of Rostock (Tierschutz-G_Nr. A414-18). Mice were decapitated under isoflurane narcosis, and the aorta was removed subsequently.

Reviewer #1 (Public review): https://doi.org/10.7554/eLife.109479.3.sa1
Reviewer #2 (Public review): https://doi.org/10.7554/eLife.109479.3.sa2
Reviewer #3 (Public review): https://doi.org/10.7554/eLife.109479.3.sa3
Author response https://doi.org/10.7554/eLife.109479.3.sa4

## Additional files

### Supplementary files

MDAR checklist

Source data 1. Summary of data used in figures and applied statistical tests with p values.

### Data availability

All mean data and statistical tests shown in figures are included as tables in source data file 1. Source data of electrophysiological measurements and molecular dockings are available from Dryad (https://doi.org/10.5061/dryad.x69p8d00j).

The following dataset was generated:

| Author(s) | Year | Dataset title | Dataset URL | Database and Identifier |
|---|---|---|---|---|
| Cordeiro S, Patejdl R, Baukrowitz T, Musinszki MA | 2026 | Data from: Natural xanthones as α-Mangostin induce vasorelaxation involving key gating residues in the S6 domain of BK channels | https://doi.org/10.5061/dryad.x69p8d00j | Dryad Digital Repository, 10.5061/dryad.x69p8d00j |

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
