## [Editor Report · eLife Assessment]

The present manuscript by Cordeiro et al. shows **convincing** evidence that α-mangostin, a xanthone obtained from the fruit of the Garcinia mangostana tree, behaves as a strong activator of the large-conductance (BK) potassium channels. The authors suggest that α-mangostin activation of the BK channel is state-independent, and molecular docking and mutagenesis suggest that α-mangostin binds to a site in the internal cavity. Additionally, the authors show that α-mangostin can relax arteries, further suggesting the plausibility of the proposed effects of this compound. These are **valuable** findings that should be of interest to channel biophysicists and physiologists alike.

---

## [Referee Report · Reviewer #1 (Public review)]

In this manuscript, the authors aimed to identify the molecular target and mechanism by which α-Mangostin, a xanthone from Garcinia mangostana, produces vasorelaxation that could explain the antihypertensive effects. Building on on prior reports of vascular relaxation and ion channel modulation, the authors convincingly show that large-conductance potassium BK channels are the primary site of action. Using electrophysiological, pharmacological, and computational evidence, the authors achieved their aims and showed that BK channels are the critical molecular determinant of mangostin's vasodiltory effects, even though the vascular studies are quite preliminary in nature.

Strengths:

(1) The broad pharmacological profiling of mangostin across potassium channel families, revealing BK channels - and the vascular BK-alpha/beta1 complex - as the potently activated target in a concentration-dependent manner.

(2) Detailed gating analyses showing large negative shifts in voltage-dependence of activation and altered activation and deactivation kinetics.

(3) High-quality single-channel recordings for open probability and dwell times.

(4) Convincing activation in reconstituted BKα/β1-Caᵥ nanodomains mimicking physiological condition and functional proof-of-concept validation in mouse aortic rings.

Weaknesses are minor:

(1) Some mutagenesis data (e.g., partial loss at L312A) could benefit from complementary structural validation.

The author's rebuttal provides alphafold3 models for mutants. While there are interesting preliminary observations, the authors decided not to include these in the main manuscript, awaiting further structual validation. I concur.

(2) While Cav-BK nanodomains were reconstituted, direct measurement of calcium signals after mangostin application onto native smooth muscle could be valuable.

In their response, the authors acknowledge the importance of measuring Ca2+ sparks in smooth muscle cells to further validate their findings. However, this is not provided in the manuscript. Part of my earlier comment alludes to the possibility of α-Mangostin directly affecting Cav1.2 or ryanodine receptor activity, and therefore BK activity would go up. With the current provided evidence, these possibilities cannot be excluded and need to be acknowledged.

(3) The work has impact for ion channel physiology and pharmacology, providing a mechanistic link between a natural product and vasodilation. Datasets include electrophysiology traces, mutagenesis scans, docking analyses, and aortic tension recordings. The latter however are preliminary in nature.

The authors acknowledge that additional vascular physiology experiments would strengthen the argument they make. They are however unable to provide such evidence in the present manuscript. Therefore, I strongly suggest that the authors tune down the physiological implications of α-Mangostin that they include in the manuscript. I'd also suggest that "vasorelaxation" is removed from the manuscript title, given the preliminary nature of the findings.

---

## [Referee Report · Reviewer #2 (Public review)]

Summary:

In the present manuscript, Cordeiro et al. show that α-mangostin, a xanthone obtained from the fruit of the Garcinia mangostana tree, behaves as an agonist of the BK channels. The authors arrive at this conclusion by examining the effects of mangostin on macroscopic and single-channel currents elicited by BK channels formed by the α subunit and α + β1 subunits, as well as αβ1 channels coexpressed with voltage-dependent Ca2+ (CaV1,2) channels. The single-channel experiments show that α-mangostin produces a robust increase in the probability of opening without affecting the single-channel conductance. The authors contend that α-mangostin activation of the BK channel is state-independent, and molecular docking and mutagenesis suggest that α-mangostin binds to a site in the internal cavity. Importantly, α-mangostin (10 μM) alleviates noradrenaline-induced contracture. Mangostin is ineffective if the contracted muscles are pretreated with the BK toxin iberiotoxin.

In this revised version of the manuscript by Cordeiro et al., the authors have adequately answered my previous concerns. However, as I stated in my comments, without determining the probability of opening across a wide range of voltages, any conclusion about the drug's mechanism of action can be questioned. For example, the statement in Discussion line 481: "The higher shift observed in 1 μM Cai 2+ may reflect the steep Cai2+-dependence of the closed-open equilibrium (Cui, Cox and Aldrich, 1997) and the allosteric coupling of voltage and Cai2+ signals (Horrigan and Aldrich, 2002; Magleby, 2003; Clay, 2017), which are effective in this concentration range, which may lead to a higher apparent activation when voltage activation is facilitated by Cai 2+ (Sun and Horrigan, 2022)." has no support in the data and is not predicted by the allosteric model. In order to have a larger shift induced by the drug in the presence of Ca2+, you need either to alter the Ca2+ binding or the allosteric coupling factor C.

Please note that in the manuscript, there are several problems with the English in this sentence.

Minor

In Figure 1E, BKa should read BKalpha.

---

## [Referee Report · Reviewer #3 (Public review)]

Summary:

This research shows that a-mangostin, a proposed nutraceutical, with cardiovascular protecting properties, could act through the activation of large conductance potassium permeable channels (BK). The authors provide convincing electrophysiological evidence that the compound binds to BK channels and induces a potent activation, increasing the magnitude of potassium currents. Since these channels are important modulators of the membrane potential of smooth muscle in vascular tissue, this activation leads to muscle relaxation, possibly explaining cardiovascular protecting effects.

Strengths:

The authors have satisfactorily answered my previous comments and present evidence based on several lines of experiments that a-mangostin is a potent activator of BK channels. The quality of the experiments and the analysis is high and represents an appropriate level of analysis. This research is timely and provides a basis to understand the physiological effects of natural compounds with proposed cardio protective effects.

Weaknesses:

The identification of the binding site continues to be the least developed point of the manuscript. The authors show that the binding site is probably located in the hydrophobic cavity of the pore and show that point mutations reduce the magnitude of the negative voltage shift of activation produces by a-mangostin. This binding site should be demonstrated in the future using structural techniques such as cryo-EM.

---

## [Author Response]

The following is the authors’ response to the original reviews.

**Public Reviews:**

**Reviewer #1 (Public review):**
In this manuscript, the authors aimed to identify the molecular target and mechanism by which α-Mangostin, a xanthone from Garcinia mangostana, produces vasorelaxation that could explain the antihypertensive effects. Building on prior reports of vascular relaxation and ion channel modulation, the authors convincingly show that large-conductance potassium BK channels are the primary site of action. Using electrophysiological, pharmacological, and computational evidence, the authors achieved their aims and showed that BK channels are the critical molecular determinant of mangostin's vasodilatory effects, even though the vascular studies are quite preliminary in nature.Strengths:(1) The broad pharmacological profiling of mangostin across potassium channel families, revealing BK channels - and the vascular BK-alpha/beta1 complex - as the potently activated target in a concentration-dependent manner.(2) Detailed gating analyses showing large negative shifts in voltage-dependence of activation and altered activation and deactivation kinetics.(3) High-quality single-channel recordings for open probability and dwell times.(4) Convincing activation in reconstituted BKα/β1-Ca_v_ nanodomains mimicking physiological conditions and functional proof-of-concept validation in mouse aortic rings.

We thank the reviewer for acknowledging the strength of the different aspects investigated in our study.

Weaknesses are minor:(1) Some mutagenesis data (e.g., partial loss at L312A) could benefit from complementary structural validation.

In the attempt to improve structural insight for the presented mutagenesis data, we have used Alphafold3 (AF3; Abramson et al., 2024) to generate models of the I308A, L312M and A316P substitutions and repeated the docking for each (Fig. R1). According to these predictive models,

The I308A substitution considerably straightens the S6 helix starting at this residue. Hence, all residues are displaced relative to the WT: C_a_ of L312, F315, and A316 are displaced by 2.8 Å, 4.2 Å, and 4.6 Å, respectively, widening the bottom of the binding pocket. However, the prediction confidence is rated lower as in the other AF3 models for all helices (70 > plDDT > 50). In the docking, poses in the binding pocket comparable to these observed in the WT (i.e. involving I308A, L312 and A316) and with the same molecule orientation have higher binding energies (-7.13 to -6.66 kcal mol^-1^). Additionally, poses without contact to I308A arise that have a more vertical position, indicating that the structural change affects the binding region.

The changes induced by L312M are localized to residues 313-323, where S6 bends towards S5. Binding energies are lower especially in the best 2 poses that are also most comparable to the WT docking (-9.88 kcal mol^-1^), but clustering overall is poor and poses are more heterogeneous. Interactions with L312M are completely abolished, while interactions with I308 (in 11/20 poses), F315 (in all poses), and A316 (in 5/20 poses) persist. Because of the rather small structural alteration induced by the substitution and the variable poses one could speculate that the reduced V_½_ shift is due to the observed loss in binding to L312M; however, retained interactions to the other residues would still allow α-Mangostin to activate.

A316P induces a displacement of the S6 helix compared to the WT while the other pore helices are not affected. S6 shows an enhanced outward bending around A316, which results in displacements of residues where a-Mangostin would bind, i.e., the C_a_ of F315 and L312M are displaced by 2.4 Å and 2.8 Å (I308 is not affected). Residues below are moved in a more rotational way, resulting in a C_a_ displacement of 3.1 Å for Y318 and even 5.7 Å for V319, before displacements decrease again towards the intracellular helix end. While interactions with A316P are present in 10/20 analyzed poses, the helix displacement seems to hinder I308 and L312 interactions, as the best docked a-Mangostin pose (-8.41 kcal mol^-1^) is predicted to only contact F315 and Y318, and overall, any I308 or L312 contacts only occurred in 3/20 and 7/20 poses (wildtype: 17/20 and 20/20 poses). This may hint at a mechanism where A316P probably has a substantial allosteric share in reducing the V_½_ shift induced by a-Mangostin and underlines the exceptional effect of this mutation (i.e., complete loss of a V_½_ shift).

**Author response image 1. sa4fig1:** Alphafold3 models of BK I308A, L312M, and A316P with α-Mangostin docked to the mutant structures. The upper row shows an overview of the mutant pore helices (AF3 models) used for molecular docking. The lower row shows the binding region with the wildtype structure overlaid in gray. Only 3 helices are shown for clarity.

Although these results provide interesting tentative explanations for the effect of the mutations and conclusions from AF3 models become increasingly robust, we think that definitive statements of their mechanistic contributions would require experimental studies of mutant channels, i.e., cryo-EM or crystallography, that are beyond our means. Therefore, we have decided not to include this data in the manuscript; however, it is accessible for the interested reader within the public review. Hopefully, as cryo-EM structures have been obtained for the wildtype channel, there will be studies on mutations of this gating-relevant S6 segment in the future.

(2) While Cav-BK nanodomains were reconstituted, direct measurement of calcium signals after mangostin application onto native smooth muscle could be valuable.

We are not sure if a global elevation of cellular calcium concentration would be informative. We rather expect that the relevant local Ca^2+^ elevation would occur as sparks in the BK-Ca_v_ nanodomains, close to the membrane. We would anticipate a change in spark duration, as the Ca^2+^ inward current would be stopped faster by the enhanced repolarization via a-Mangostin activated BKα/β1 channels. This would require fast Ca^2+^ imaging acquisition speed to capture spark activity. We concur that this would be an informative experiment to investigate a more native situation. However, we would have to accomplish such methodologically challenging measurements in a separate project, which could fruitfully be combined with a more extensive characterization of aortic contraction as also suggested in the following remark (3).

(3) The work has an impact on ion channel physiology and pharmacology, providing a mechanistic link between a natural product and vasodilation. Datasets include electrophysiology traces, mutagenesis scans, docking analyses, and aortic tension recordings. The latter, however, are preliminary in nature.

We completely agree with the reviewer that there is ample room for further studies that could characterize different tissues important in blood pressure regulation (such as resistance arteries), elucidate even more physiological detail (such as modulatory effects of the endothelium), or look deeper into the pharmacology using chemically altered Mangostin derivatives. While we very much like this to happen in future projects, in this study we focused on the functional aspects of a-Mangostin in BK channel gating. We present our tension recordings as a proof-of-concept to underline the activity of a-Mangostin in native tissues, and we clearly show the importance of the BK channel by using iberiotoxin as a specific inhibitor which impressively abolished relaxation.

References:

Abramson, J. et al. (2024) “Accurate structure prediction of biomolecular interactions with AlphaFold 3,” Nature, 630(8016), pp. 493–500. Available at: https://doi.org/10.1038/s41586-024-07487-w.

**Reviewer #2 (Public review):**
Summary:In the present manuscript, Cordeiro et al. show that α-mangostin, a xanthone obtained from the fruit of the Garcinia mangostana tree, behaves as an agonist of the BK channels. The authors arrive at this conclusion through the effect of mangostin on macroscopic and single-channel currents elicited by BK channels formed by the α subunit and α + β1 sununits, as well as αβ1 channels coexpressed with voltage-dependent Ca2+ (CaV1,2) channels. The single-channel experiments show that α-mangostin produces a robust increase in the probability of opening without affecting the single-channel conductance. The authors contend that α-mangostin activation of the BK channel is state-independent and molecular docking and mutagenesis suggest that α-mangostin binds to a site in the internal cavity. Importantly, α-mangostin (10 μM) alleviates the contracture promoted by noradrenaline. Mangostin is ineffective if the contracted muscles are pretreated with the BK toxin iberiotoxin.Strengths:The set of results combining electrophysiological measurements, mutagenesis, and molecular docking reveals α-mangostin as a potent activator of BK channels and the putative location of the α-mangostin binding site. Moreover, experiments conducted on aortic preparations from mice suggest that α-mangostin can aid in developing drugs to treat a myriad of diverse diseases involving the BK channel.

We thank the reviewer for pointing out the significance of our study.

Weaknesses:Major:(1) Although the results indicate that α-mangostin is modifying the closed-open equilibrium, the conclusion that this can be due to a stabilization of the voltage sensor in its active configuration may prove to be wrong. It is more probable that, as has been demonstrated for other activators, the α-mangostin is increasing the equilibrium constant that defines the closed-open reaction (L in the Horrigan, Aldrich allosteric gating model for BK). The paper will gain much if the authors determine the probability of opening in a wide range of voltages, to determine how the drug is affecting (or not), the channel voltage dependence, the coupling between the voltage sensor and the pore, and the closed-open equilibrium (L).

We would like to take the opportunity to clarify this potential misunderstanding. In our manuscript, we have discussed three mechanistic explanations for the Mangostin activation: (1) an electrostatic effect at the selectivity filter, (2) structural and electrostatic changes of S6 that facilitate the opening of a putative lower gate, and (3) hydrophobic gating, i.e., counteracting dewetting of the pore. All possibilities would impact S6 and lower the free energy for pore opening, and we concur that therefore Mangostin most likely affects the closed-open equilibrium (L) of the BKα channel.

The sentence at the original lines 470-471, “(…) caused by an enhanced shift of the closed-open equilibrium toward the open state, such as the stabilization of the voltage sensor in an active conformation” refers to the observation that the presence of the β1 subunit enhances this closed-open shift. The stabilization of the voltage sensor domain was mentioned as one example of how it achieves this. We recognize that this example was an unfortunate choice, as β1 rather facilitates Ca^2+^-dependent allosteric pore opening unrelated to the discussed mechanisms of Mangostin. We have therefore removed this statement.

As to the suggestion to dissect the effect of Mangostin on C, D, and L, we agree with the reviewer that this would surely add to a full biophysical characterization. However, in our project, we strove towards including more experiments showing the physiological implications of Mangostin activation to emphasize the implication for vasodilation. We hope the reviewer understands that, with limited resources, this came at the expense of a full investigation of the different gating components, which could pose a separate project by itself.

(2) Apparently, the molecular docking was performed using the truncated structure of the human BK channel. However, it is unclear which one, since the PDB ID given in the Methods (6vg3), according to what I could find, corresponds to the unliganded, inactive PTK7 kinase domain. Be as it may, the apo and Ca2+ bound structures show that there is a rotation and a displacement of the S6 transmembrane domain. Therefore, the positions of the residues I308, L312, and A316 in the closed and open configurations of the BK channel are not the same. Hence, it is expected that the strength of binding will be different whether the channel is closed or open. This point needs to be discussed.

We apologize for the typing error and thank the reviewer for indicating this erroneous PDB ID. (“6vg3”). It should have read PDB ID 6v3g as in the legend to Fig. 4B. The reviewer appropriately points out that there are differences in the S6 segment addressed in our study between the two available cryo-EM structures obtained in the presence (PDB ID 6v38) and absence of Ca^2+^ (PDB ID 6v3g) (Tao and MacKinnon, 2019).

We had actually performed the docking with both structures, but chosen to show the Ca^2+^-free structure to better visualize the I308 position. a-Mangostin is found in the same S6 region in both, not obstructing the K^+^ conduction pathway. The binding energies of the favored poses are very similar; the binding energy in the best-ranking conformational cluster in the Ca^2+^-bound structure even was slightly lower (-8.64 kcal mol^-1^) than in the docking with the Ca^2+^-free channel (-8.58 kcal mol^-1^; Fig. 4B), which may not be a relevant difference.

We compared the residue interactions in both dockings (Author response table 1). S317 and Y318, which did not reduce the shift in V_½_ upon substitution, were not predicted to contact a-Mangostin in either structure. In both structures, L312 and F315 were predicted to interact in virtually all poses analyzed. In the docking to the Ca^2+^-free state, also I308 was predicted to interact in 17/20 poses, while contacts to A316 occurred in 5/20 poses. In the Ca^2+^-bound state, predicted interactions shifted from I308 (which is expected as it is buried in the protein) to A316, and the isoprenyl moiety close to I308 rotated downwards. This could indicate that a-Mangostin adopts a more horizontal position following the upward reorientation of S6 in the Ca^2+^-bound state when the channel moves from one to the other conformation (Fig. S4).

**Author response table 1. sa4table1:** Number of interactions of S6 residues in 20 analyzed α-Mangostin poses in the molecular dockings to the Ca2+-free and Ca2.

Residue	Ca^(2+)-free structure (6v3g)	Ca^(2+)-bound structure (6v38)
1308	17/20 poses	0/20 poses
L312	20/20	20/20
F315	20/20	19/20
A316	5/20	19/20
S317	0/20	0/20
Y318	0/20	0/20

These docking results are consistent with our functional measurements. Recent structures of the BK/γ1 complex showed that the VSD and Ca^2+^-bowl are stabilized in an active-like conformation that corresponds to the conformation seen in the Ca^2+^-bound state (Kallure et al., 2023; Yamanouchi et al., 2023; Redhardt, Raunser and Raisch, 2024), indicating that very likely the Ca^2+^-bound and Ca^2+^-free structures indeed represent open and closed conformations of the channel. We observed that α-Mangostin can bind to both of these states to activate the channel (Fig. 3C, D), showing the presence of a binding site in both conformations. Further, α-Mangostin induced a left-shift in V_½_ also in higher Ca^2+^ concentration (Fig. 2D), indicating that it still binds to and activates the channel after the conformational change in S6. As we could not determine affinity for the mutants due to limited solubility, we have no information on the nature of the contribution of the substitutions, i.e., reduced binding or allosteric effect. As I308 is buried in the Ca^2+^-bound state, its contribution is likely mostly allosteric. We have also proposed dewetting as possible activation mechanism, which we expect to be less sensitive to the exact pose of a molecule (as shown for NS11021, Nordquist et al., 2024). Therefore, α-Mangostin could, e.g., change solvent accessibility of the I308 sidechain, energetically favoring the buried (open) state.

We have now included both dockings and Author response table 1 in Fig. S4, and we have added passages to the results section (starting at line 373) and discussion section (starting at lines 544, 588).

Minor:(1) From Figure 3A, it is apparent that the increase in Po is at the expense of the long periods (seconds) that the channel remains closed. One might suggest that α-mangostin increases the burst periods. It would be beneficial if the authors measured both closed and open dwell times to test whether α-mangostin primarily affects the burst periods.

We thank the reviewer for this valuable suggestion, which we have implemented. In our single channel measurements shown in our original Fig. 3 we have not observed burst behavior of the BKɑ channels. This can be explained by the fact that we measured in resting condition (100 nM free Ca_i_2+) and with rather mild depolarisation (+40 mV) where Po was very low. We have therefore analyzed measurements in 5 µM free a_i_2+ where we recorded sufficient burst activity also in the basal state.

The burst analysis showed that ɑ-Mangostin indeed prolongs bursts and shortens the interburst closures. Within bursts, both closed times and open times were increased, and we recorded a higher number of opening events per burst. We conclude that ɑ-Mangostin acts in both the closed and the open state, where it slows open-closed transitions resulting in less flicker, and stabilizes the open state via longer open times and a higher probability for closed-open transitions.

We now show this data in Fig. 3D-F and Table S8, and have accordingly added passages to the results section (starting at line 285), the discussion (line 510), and the methods section (starting at line 746).

(2) In several places, the authors make similarities in the mode of action of other BK activators and α-mangostin; however, the work of Gessner et al. PNAS 2012 indicates that NS1619 and Cym04 interact with the S6/RCK linker, and Webb et al. demonstrated that GoSlo-SR-5-6 agonist activity is abolished when residues in the S4/S5 linker and in the S6C region are mutated. These findings indicate that binding of the agonist is not near the selectivity filter, as the authors' results suggest that α-mangostin binds.

We will gladly clarify our ideas concerning the binding sites of other activators and ɑ-Mangostin. We first hypothesized that ɑ-Mangostin may share characteristics and mode of action with the class of negatively charged activators (NCA) that we have described before (Schewe et al., 2019). NCA were found to occupy a common fenestration site that is located close to the selectivity filter in TREK K2P channels, and in this manuscript we have shown by THexA competition and mutagenesis experiments that ɑ-Mangostin also binds in this fenestration region in TREK-1 channels (Fig. S3).

The existence of this common NCA binding site was also proposed for BK channels, as a docking placed the NCA NS11021 in an equivalent binding region, and, among others, NS11021 and GoSlo-SR-5-6 competed with THexA for binding in the pore (Schewe et al., 2019). These results were indeed not fully in agreement with the proposed binding site of GoSlo-SR-5-6 in Webb et al. (2015), although the most effective (double) mutants were located at S317 and I323, at the intracellular end of the cleft between neighboring S6 segments. In this manuscript, we have shown that α-Mangostin is present in the pore of BK channels by molecular docking, a THexA competition assay, and two mutations that reduced the shift in V_½_ induced not only by ɑ-Mangostin but also by GoSlo-SR-5-6 (Fig. 4). While the docking was rather a starting point, both functional tests argue against a binding site in the S4/5 linker/S6C region; however, allosteric mechanisms could still reduce activation also in mutants in the S4/5 linker/S6C region far from the pore binding region proposed by us in the 2019 study and the present manuscript.

To summarize, we did not mean to imply that all BK activators should bind to this site, especially if they are not part of the NCA class (as NS1619, Cym4, as well as BC5, whose different binding site enabled us to use it as a control in our THexA competition assay). However, the cleft close to gating relevant S6 residues may well pose a region especially susceptible to modulator binding (as BL-1249, GoSlo-SR-5-6, and ɑ-Mangostin). We have moved, respectively separated, the initial GoSlo references from the reference to the pore binding site in the paragraph (lines329, 358) to improve clarity.

(3) The sentence starting in line 452 states that there is a pronounced allosteric coupling between the voltage sensors and Ca2+ binding. If the authors are referring to the coupling factor E in the Horrigan-Aldrich gating model, the references cited, in particular, Sun and Horrigan, concluded that the coupling between those sensors is weak.

We are grateful for the opportunity to improve this passage. We intended to express that observed effects (in this case the shift in V_½_) are pronounced around 1 µM Ca^2+^. As the reviewer states, the coupling factor between the voltage and calcium sensors (E; 2.4) is weak compared to the coupling of Ca^2+^ (C; 8) and voltage (D; 25) to the pore in the Horrigan-Aldrich model. However, the shape of the Ca^2+^-dependence of V_½_ cannot be completely described when E is neglected, with the highest difference around 1-2 µM Ca^2+^ (Horrigan and Aldrich, 2002). Deletion of the gating ring underlines the allosteric sensor coupling (Clay, 2017). This together with the steep Ca^2+^-dependence in this concentration range (meaning high Po changes upon occupancy increase; Cui, Cox and Aldrich, 1997) explains the higher apparent activation, visible as the higher shift in V_½_ observed at the 1 µM Ca^2+^. Speaking with the model of Sun and Horrigan (2022), the suppressing “molecular logic gate” is already relieved by the presence of intermediate Ca^2+^, and the direct “gating lever” pathway via voltage acts synergistically and achieves the observed higher V_½_ shift upon depolarization. We have adapted the sentence and separated the citations for better understanding (lines 503-507).

References:

Clay, J.R. (2017) “Novel description of the large conductance Ca2+-modulated K+ channel current, BK, during an action potential from suprachiasmatic nucleus neurons,” Physiological Reports, 5(20), p. e13473. Available at: https://doi.org/10.14814/phy2.13473.

Cui, J., Cox, D.H. and Aldrich, R.W. (1997) “Intrinsic Voltage Dependence and Ca2+ Regulation of mslo Large Conductance Ca-activated K+ Channels,” Journal of General Physiology, 109(5), pp. 647–673. Available at: https://doi.org/10.1085/jgp.109.5.647.

Horrigan, F.T. and Aldrich, R.W. (2002) “Coupling between voltage sensor activation, Ca2+ binding and channel opening in large conductance (BK) potassium channels,” The Journal of General Physiology, 120(3), pp. 267–305. Available at: https://doi.org/10.1085/jgp.20028605.

Kallure, G.S. et al. (2023) “High-resolution structures illuminate key principles underlying voltage and LRRC26 regulation of Slo1 channels.” bioRxiv, p. 2023.12.20.572542. Available at: https://doi.org/10.1101/2023.12.20.572542.

Nordquist, E.B., Jia, Z., Chen, J., 2024. “Small Molecule NS11021 Promotes BK Channel Activation by Increasing Inner Pore Hydration.” J. Chem. Inf. Model. 64, 7616–7625. https://doi.org/10.1021/acs.jcim.4c01012

Redhardt, M., Raunser, S. and Raisch, T. (2024) “Cryo-EM structure of the Slo1 potassium channel with the auxiliary γ1 subunit suggests a mechanism for depolarization-independent activation,” FEBS Letters, 598(8), pp. 875–888. Available at: https://doi.org/10.1002/1873-3468.14863.

Schewe, M. et al. (2019) “A pharmacological master key mechanism that unlocks the selectivity filter gate in K + channels.,” Science, 363(6429), pp. 875–880. Available at: https://doi.org/10.1126/science.aav0569.

Sun, L. and Horrigan, F.T. (2022) “A gating lever and molecular logic gate that couple voltage and calcium sensor activation to opening in BK potassium channels,” Science Advances, 8(50), p. eabq5772. Available at: https://doi.org/10.1126/sciadv.abq5772.

Tao, X. and MacKinnon, R. (2019) “Molecular structures of the human Slo1 K+ channel in complex with β4,” eLife 8, p. e51409. Available at: https://doi.org/10.7554/eLife.51409.

Webb, T.I. et al. (2015) “Molecular mechanisms underlying the effect of the novel BK channel opener GoSlo: Involvement of the S4/S5 linker and the S6 segment,” Proceedings of the National Academy of Sciences, 112(7), pp. 2064–2069. Available at: https://doi.org/10.1073/pnas.1400555112.

Yamanouchi, D. et al. (2023) “Dual allosteric modulation of voltage and calcium sensitivities of the Slo1-LRRC channel complex,” Molecular Cell, 83(24), pp. 4555-4569.e4. Available at: https://doi.org/10.1016/j.molcel.2023.11.005.

**Reviewer #3 (Public review):**
Summary:This research shows that a-mangostin, a proposed nutraceutical, with cardiovascular protective properties, could act through the activation of large conductance potassium permeable channels (BK). The authors provide convincing electrophysiological evidence that the compound binds to BK channels and induces a potent activation, increasing the magnitude of potassium currents. Since these channels are important modulators of the membrane potential of smooth muscle in vascular tissue, this activation leads to muscle relaxation, possibly explaining cardiovascular protective effects.Strengths:The authors present evidence based on several lines of experiments that a-mangostin is a potent activator of BK channels. The quality of the experiments and the analysis is high and represents an appropriate level of analysis. This research is timely and provides a basis to understand the physiological effects of natural compounds with proposed cardio-protective effects.

We sincerely thank the reviewer for appraising the achievements of our study.

Weaknesses:The identification of the binding site is not the strongest point of the manuscript. The authors show that the binding site is probably located in the hydrophobic cavity of the pore and show that point mutations reduce the magnitude of the negative voltage shift of activation produced by a-mangostin. However, these experiments do not demonstrate binding to these sites, and could be explained by allosteric effects on gating induced by the mutations themselves.

We are aware that our functional data are unfortunately not sufficient to clearly distinguish between effects due to affinity loss or due to allosteric mechanisms. Our attempts to generate complete dose–response curves for the mutants to determine accurate apparent IC_50_ values were unfortunately limited by the solubility of the compound. Consequently, we have avoided making claims about affinity loss in the mutant analysis, and have instead only reported the reduction in potency, expressed as the shift in V_½_. To reduce confounding effects from the mutations themselves, we selected substitutions that preserved the most wildtype-like GV-relationships, based on the extensive mutagenesis work of (Chen, Yan and Aldrich, 2014). We address this matter also in our answer to Recommendation (6) below, and we have replaced the word “binding” in the title of the manuscript. Nevertheless, we consider the proposed binding region to be well supported by the THexA competition experiments in combination with molecular docking, even though the specific mechanistic contributions of individual residues cannot yet be resolved.

**Reviewer #3 (Recommendations for the authors):**
(1) Natural xanthones as α-Mangostin induce vasorelaxation via binding to key gating residues in the S6 domain of BK channels.(2) If α-Mangostin occupies a similar binding site to quaternary ammoniums, what is the explanation for not observing a reduction in the single-channel current (fast blocking effect)? The α-Mangostin site proposed here is in a region of the channel that should occlude ion permeation. The authors should discuss possible explanations for this apparently contradictory observation.

As the reviewer states, we indeed have not observed a reduced single channel amplitude in any measurement. The THexA competition assay showed that ɑ-Mangostin is present in the pore cavity and interferes with THexA access to its binding site. However, we do not think that their binding sites are similar, as QA ions bind directly below the filter entrance to block permeation, while our studies suggest that ɑ-Mangostin binds in the upper portion of the cleft between S6 helices. In this position, it would clearly overlap with the QA binding site and hinder access, but not block permeation. We would therefore not expect to see an amplitude reduction by intermittent α-Mangostin block. Consistently, all binding poses in our dockings were close to the cavity wall, without interfering with the central ion conduction pathway. To better illustrate this, we have added updated intracellular views of the dockings in the Ca^2+^-free and Ca^2+^-bound state (which we have also now included as suggested by another reviewer) to the supplementary information (Fig. S4A).

(3) In Figure 2D, it is difficult to appreciate the differences between the symbols representing the G-V relationships of BKa channels at different intracellular Ca concentrations, before and after activation with 10 μM a-Mangostin. A clearer distinction between the curves would help to interpret the data more easily.

We thank the reviewer for the suggestion to improve figure accessibility. We have changed the line appearance for better discrimination of the overlying portions.

(4) Both THexA and TPA block BK channels through voltage and state-dependent mechanisms. Therefore, their apparent affinity could change if a-Mangostin simply increases open probability or alters dwell times rather than physically blocking access to the binding site.

The reviewer addresses valid limitations that can affect the meaningfulness of competition experiments under certain conditions. However, we think that this does not apply to our results:

Previous studies have shown that the voltage dependence of quaternary ammonium blockers up to C_10_ is rather weak in BK channels, and only a slight increase in block is present in the voltage range +30 mV to +100 mV (Li and Aldrich, 2004; Thompson and Begenisich, 2012). Hence, THexA voltage dependence has already reached a plateau in the competition assay (at +40 mV), and its voltage dependence would have little effect on our results.

Controversy exists about the nature of the state dependence of different quaternary ammonium blockers, but TBA is often recognized as an open channel blocker of BK channels, which probably also applies to THexA (Wilkens and Aldrich, 2006; Tang, Zeng and Lingle, 2009; Thompson and Begenisich, 2012; Posson, McCoy and Nimigean, 2013). Assuming such an open-channel block, apparent IC_50_ values would be inversely proportional to Po. The THexA IC_50_ was about 80 nM in the basal state, when Po is very low (0.024 at +40 mV as derived from the GV-relationship); an increase of open dwell times, respectively Po, in the presence of α-Mangostin to, e.g., 0.3 would therefore lead to a ≈10-fold decrease in apparent IC_50_. However, the apparent THexA IC_50_ strongly increased rather than decreased (more than 20-fold to around 1.6 µM). This cannot arise from Po change and must reflect the altered access of THexA to its binding site caused by α-Mangostin. Assuming a pure closed channel block where apparent IC_50_ would correlate with the closed times, an increase of about 1.4-fold is expected. However, we recorded a much stronger 20-fold increase. Therefore, we are convinced that we have conclusively shown that α-Mangostin is present in the BK pore irrespective of the state dependence of THexA block.

(5) The pH dependence of the V1/2 shift supports the idea that α-Mangostin becomes more negatively charged at higher pH (enhancing its effect.) However, although the data are consistent with this interpretation, additional controls such as using a non-ionizable analog or assessing solubility changes with pH would be needed to confirm that the shift is caused specifically by ionization of α-Mangostin and not by indirect pH effects on channel gating.

We agree with the reviewer that the pH experiment by itself is not sufficient to clearly tie the existence of a charge to a possible activation mechanism. We still think that this is an interesting observation and should be made known, as we have investigated the mechanism of negatively charged activators in different K^+^ channel families before (Schewe et al., 2019). Unfortunately, we do not have access to uncharged derivatives mimicking the 3D conformation. From the commercially available substances, the bare xanthone backbone is completely insoluble in water. We have therefore tested the derivative 3-hydroxyxanthone as example with a minimal number of hydroxyl substituents (Author response image 2, Author response table 2). The 3-hydroxyxanthone indeed shows reduced activation compared to α-Mangostin. The shift in V_½_ induced by 10 µM 3-hydroxyxanthone was only 14.99 ± 5.67 mV (≈50 mV for α-Mangostin). This supports that the presence of several (potentially) charged substituents is important for the activation mechanism. However, we have no knowledge about the efficacy of the compound or the local pK_a_ of the different hydroxyl groups. As the reviewer stated, systematic chemical modifications would be necessary to elucidate the importance of the charged substituent number and positions, which is not within our capabilities.

**Author response image 2. sa4fig2:** Activation of BKα by 3-hydroxyxanthone. (A) GV-relationship before and after application of 10 µM 3-hydroxyxanthone. (B) V_½_ before and after application of 10 µM 3-hydroxyxanthone compared to α-Mangostin and the resulting difference in V_½_ (ΔV_½_). Measurements were conducted as described in the main manuscript with 100 nM free Ca_i_^2+^.

**Author response table 2. sa4table2:** Comparison of the V_½_ ± SEM and ΔV_½_ ± SEM before and after activation by 10 µM α-Mangostin or 10 µM 3-hydroxyxanthone in BKα channels. Unpaired t-test, two-tailed P values (α=0.05)

	V_(1//2)(mV) basal	V_(1//2)(mV) activated	DeltaV_(1//2)(mV)	P-value
alpha-Mangostin	110.49+-2.69	57.37+-3.60	53.08+-4.9	< 0.001
3-hydroxyxanthone	111.82+-3.93	96.83+-1.01	14.99+-5.67	

(6) The reduced V1/2 shifts observed in the I308A, L312M, and A316PP mutants may result from intrinsic gating alterations rather than a true loss of a-Mangostin binding. The GoSlo-SR-5-6 control is informative, but the persistence of activation in A316P does not fully resolve this. A more convincing test would be employing double or triple mutants.

As stated above, we acknowledge that our functional data do not allow us to definitively separate effects arising from a true loss of binding affinity from those due to potential allosteric effects. We tried to minimize intrinsic gating alteration brought by substitutions by not conducting a pure alanine or cysteine scanning mutagenesis. Instead, substitutions were chosen to be closest to the wildtype GV-relationship in (Chen, Yan and Aldrich, 2014) where possible. While L312M was virtually identical to the wildtype, A316P showed a change in slope in high Ca^2+^ concentrations, which could indicate a changed voltage sensitivity. Additionally, A316P completely abolished α-mangostin activation. We therefore also used A316G to ensure that the channel is functional and retains voltage sensitivity, even if its V_½_ was shifted stronger. As we have conducted paired measurements and assessed the V_½_ before and after activation, we are confident that we can attribute a reduced shift to the reduced action of α-mangostin.

Following the reviewer’s suggestion, we have generated and measured the double mutants I308A/L312M, I308A/A316G, and L312M/A316G (the triple mutant I308A/L312M/A316G did not produce measurable currents). The mutants I308A/L312M and I308A/A316G showed a moderate energy-additive effect and reduced the shift in V_½_ by further ≈7 mV compared to the single mutation with the stronger shift. The combination L312M/A316G, however, did not further reduce the shift seen in the single mutations and did not even produce the shift induced by A316G alone.

**Author response image 3. sa4fig3:** Double Mutants I308A/L312M, I308A/A316G and L312M/A316G compared to the single mutations in the main manuscript. The V½ before and after activation with 10 µM α-Mangostin, the resulting shift in V½, and the GV-relationships are shown (n=6-7), measurements were made as in Fig. 4.

**Author response table 3. sa4table3:** Summary of the V_½_ before and after Mangostin activation and the resulting shifts in V_½_ for the double mutants compared to the single mutants shown in the main manuscript.

	V_((1)/(2))(mV)	V_(1//2)(mV) in 10muMalpha-Mangostin	DeltaV_((1)/(2))(mV)
1308A	54.45+-3.86	34.48+-3.81	19.97+-3.12
L312M	126.29+-6.19	98.39+-4.46	27.89+-5.42
A316G	36.7+-2.96	13.13+-3.11	23.57+-2.05
1308A/L312M	60.57+-3.77	47.88+-0.77	12.7+-3.37
I308A/A316G	30.98+-1.94	14.47+-2.83	16.50+-1.48
L312M/A316G	51.55+-1.99	22.69+-5.81	28.86+-6.65

Following a suggestion by another reviewer, we have generated Alphafold3 (AF3) models for I308A, L312M and A316P and repeated the Mangostin docking. We learned that the mutations are all predicted to substantially impact the structure of the S6 helix, therefore altering the binding region, and A316P especially impacted the nature of residue interactions. This could be an explanation why the double mutants do not show a clear and consistent additive effect.

Unfortunately, this outcome is not conclusive and the double mutants do not reveal further information compared to the single mutants. We have therefore decided not to include these measurements in the manuscript.

As we do not know if our answers will be sent to all reviewers, we repeat the relevant part about the AF3 models here:

(…) According to these predictive models,

The I308A substitution considerably straightens the S6 helix starting at this residue. Hence, all residues are displaced relative to the WT: C_a_ of L312, F315, and A316 are displaced by 2.8 Å, 4.2 Å, and 4.6 Å, respectively, widening the bottom of the binding pocket. However, the prediction confidence is rated lower as in the other AF3 models for all helices (70 > plDDT > 50). In the docking, poses in the binding pocket comparable to these observed in the WT (i.e. involving I308A, L312 and A316) and with the same molecule orientation have higher binding energies (-7.13 to -6.66 kcal mol^-1^). Additionally, poses without contact to I308A arise that have a more vertical position, indicating that the structural change affects the binding region.

The changes induced by L312M are localized to residues 313-323, where S6 bends towards S5. Binding energies are lower especially in the best 2 poses that are also most comparable to the WT docking (-9.88 kcal mol^-1^), but clustering overall is poor and poses are more heterogeneous. Interactions with L312M are completely abolished, while interactions with I308 (in 11/20 poses), F315 (in all poses), and A316 (in 5/20 poses) persist. Because of the rather small structural alteration induced by the substitution and the variable poses one could speculate that the reduced V_½_ shift is due to the observed loss in binding to L312M; however, retained interactions to the other residues would still allow α-Mangostin to activate.

A316P induces a displacement of the S6 helix compared to the WT while the other pore helices are not affected. S6 shows an enhanced outward bending around A316, which results in displacements of residues where a-Mangostin would bind, i.e., the C_a_ of F315 and L312M are displaced by 2.4 Å and 2.8 Å (I308 is not affected). Residues below are moved in a more rotational way, resulting in a C_a_ displacement of 3.1 Å for Y318 and even 5.7 Å for V319, before displacements decrease again towards the intracellular helix end. While interactions with A316P are present in 10/20 analyzed poses, the helix displacement seems to hinder I308 and L312 interactions, as the best docked a-Mangostin pose (-8.41 kcal mol^-1^) is predicted to only contact F315 and Y318, and overall, any I308 or L312 contacts only occurred in 3/20 and 7/20 poses (wildtype: 17/20 and 20/20 poses). This may hint at a mechanism where A316P probably has a substantial allosteric share in reducing the V_½_ shift induced by a-Mangostin and underlines the exceptional effect of this mutation (i.e., complete loss of a V_½_ shift). (…)

(7) The subtraction approach used to isolate BK currents (difference before and after a-Mangostin) assumes that the compound affects only BK channels. However, a-Mangostin could also modulate Cav currents directly, as reported for other polyphenolic compounds. No vehicle (DMSO) control is shown.

We agree with the reviewer that α-Mangostin could also modulate Ca_v_ currents; however, this would not interfere with the conclusions drawn from this nanodomain experiment. We intended to show the overall current modulation by ɑ-Mangostin in the voltage range relevant for Ca_v_-BK coupling, as this would be the determinant for the membrane potential mediating the vasoactive effect. In native tissue, BK and Ca_v_ channels (among others) would likewise contribute to the net membrane conductance, with BK channels being a major contributor when activated. In fact, a concomitant inhibition of Ca_v_ channels could act synergistically in favor of vasodilation. This could therefore be a subject for the further investigation of potential ɑ-Mangostin targets. However, the fact that iberiotoxin prevented relaxation in aortic preparations conclusively showed that BK channels are the major player in native tissue.

We have reformulated some sentences to prevent misunderstandings that we refer to isolated BK currents instead of α-Mangostin activated currents.

DMSO controls were conducted and did not impact BK or Ca_v_1.2 currents or the aortic tissue contraction. We have added representative measurements as Fig. S6 and stated the DMSO concentration in the Methods section (line 655).

(8) Most kinetic fits were obtained at strong depolarizations (around +100 mV), which limits how well these results can be extrapolated to physiological voltages. Although the BK-Cav experiments show facilitation between -50 and +50 mV, providing plots for activation and deactivation in that range would strengthen the physiological relevance.

We thank the reviewer for this valuable suggestion. We now additionally show that the impact of ɑ-Mangostin on activation is high at lower depolarisation, indeed underlining its physiological relevance. To address the activation time course in a more physiological voltage range, we have used our measurements of BKɑ channels in 10 µM Ca_i_2+ (where the V_½_ shift induced by ɑ-Mangostin is equal to 100 nM ca_i_^2+^+; Fig. 2D). The outward currents already present in the lower voltage range under these conditions allowed us to fit a monoexponential function to the traces of 0 mV to 100 mV prepulses. The τ of activation decreased from 29.6 ± 3.1 ms at 0 mV to 2.4 ± 2 ms at +100 mV. After ɑ-Mangostin activation, the time course was accelerated, with a τ of activation of 9.5 ± 4.7 ms at 0 mV to 2 ± 0.6 ms at +100 mV. This faster activation was particularly effective in the lower voltage range far from high Po, e.g., ɑ-Mangostin caused a decrease of more than half of the τ of activation at +20 mV (from 12.2 ± 0.6 ms to 4.98 ± 1.6 ms).

Our data consists of families of different prepulse voltages and a fixed repolarisation step (to -50 mV for 100 nM free Ca_i_^2+^, and to -100 mV for 10 µM free Ca_i_^2+^). Thus, we are not able to add plots for the voltage-dependence of deactivation in the same way as for activation. However, we can present the deactivation time constants of lower prepulse voltage steps that produce outward currents in symmetrical ion conditions with 10 µM free Ca_i_2+. For -20 mV and +20 mV prepulse voltages, which better reflect physiological depolarisation, the deactivation time constant shows a 3-to 5-fold increase after ɑ-Mangostin activation.

We now show the plot for the voltage dependence of activation in Fig. S2A and a bar graph for activation/ deactivation time constants at +20 mV as Fig. S2B; data are summarized in Table S5. We hope this adds to illustrating the effect of ɑ-Mangostin under physiological conditions.

(9) Minor: In several parts of the paper, induced shifts to negative voltages are referred to "leftward shifts". It would be useful to be consistent and employ a more specific reference to negative or positive directions.

We thank the reviewer for the careful reading and have harmonized the terminology.

References

Chen, X., Yan, J. and Aldrich, R.W. (2014) “BK channel opening involves side-chain reorientation of multiple deep-pore residues,” Proceedings of the National Academy of Sciences, 111(1), pp. E79–E88. Available at: https://doi.org/10.1073/pnas.1321697111.

Li, W. and Aldrich, R.W. (2004) “Unique Inner Pore Properties of BK Channels Revealed by Quaternary Ammonium Block,” Journal of General Physiology, 124(1), pp. 43–57. Available at: https://doi.org/10.1085/jgp.200409067.

Posson, D.J., McCoy, J.G. and Nimigean, C.M. (2013) “The voltage-dependent gate in MthK potassium channels is located at the selectivity filter,” Nature Structural & Molecular Biology, 20(2), pp. 159–166. Available at: https://doi.org/10.1038/nsmb.2473.

Schewe, M. et al. (2019) “A pharmacological master key mechanism that unlocks the selectivity filter gate in K + channels.,” Science, 363(6429), pp. 875–880. Available at: https://doi.org/10.1126/science.aav0569.

Tang, Q.-Y., Zeng, X.-H. and Lingle, C.J. (2009) “Closed-channel block of BK potassium channels by bbTBA requires partial activation,” The Journal of General Physiology, 134(5), pp. 409–436. Available at: https://doi.org/10.1085/jgp.200910251.

Thompson, J. and Begenisich, T. (2012) “Selectivity filter gating in large-conductance Ca2+-activated K+ channels,” Journal of General Physiology, 139(3), pp. 235–244. Available at: https://doi.org/10.1085/jgp.201110748.

Wilkens, C.M. and Aldrich, R.W. (2006) “State-independent block of BK channels by an intracellular quaternary ammonium.,” The Journal of General Physiology, 128(3), pp. 347–364. Available at: https://doi.org/10.1085/jgp.200609579.